# VISTA checkpoint inhibition by pH-selective antibody SNS-101 with optimized safety and pharmacokinetic profiles enhances PD-1 response

Thomas Thisted[1], F. Donelson Smith [1], Arnab Mukherjee[1], Yuliya Kleschenko[1], Feng Feng[1], Zhi-Gang Jiang [1], Timothy Eitas[1], Kanam Malhotra[1], Zuzana Biesova[1], Adejumoke Onumajuru [1], Faith Finley[1], Anokhi Cifuentes[1], Guolin Zhang[1], Gaëlle H. Martin[2], Yoshiko Takeuchi [3], Kader Thiam[2], Robert D. Schreiber[3] & Edward H. van der Horst [1] ✉

VISTA, an inhibitory myeloid-T-cell checkpoint, holds promise as a target for cancer immunotherapy. However, its effective targeting has been impeded by issues such as rapid clearance and cytokine release syndrome observed with previous VISTA antibodies. Here we demonstrate that SNS-101, a newly developed pH-selective VISTA antibody, addresses these challenges. Structural and biochemical analyses confirmed the pH-selectivity and unique epitope targeted by SNS-101. These properties confer favorable pharmacokinetic and safety profiles on SNS-101. In syngeneic tumor models utilizing human VISTA knock-in mice, SNS-101 shows in vivo efficacy when combined with a PD-1 inhibitor, modulates cytokine and chemokine signaling, and alters the tumor microenvironment. In summary, SNS-101, currently in Phase I clinical trials, emerges as a promising therapeutic biologic for a wide range of patients whose cancer is refractory to current immunotherapy regimens.

Immunotherapies, particularly immune checkpoint inhibitors, have revolutionized cancer treatment. Despite the remarkable clinical responses from blocking the programmed cell death protein 1 (PD-1)/ programmed cell death-ligand 1 (PD-L1) axis across a spectrum of indications, resistance to anti-PD-1 therapy remains a significant challenge[1]. This underscores the ongoing need for novel drugs that target diverse immune regulators. VISTA (V-domain immunoglobulin (Ig) suppressor of T-cell activation), a B7 family member, promotes T-cell and myeloid quiescence and represents a promising target, especially in combination with anti-PD-1/PD-L1 treatment[2–6]. Recent studies elucidated the pH-dependent interaction between VISTA and the T-cell checkpoint receptor, P-selectin glycoprotein ligand-1 (PSGL-1)[7]. This interaction is

particularly relevant in the acidic tumor microenvironment (TME), characterized by extracellular acidosis due to deregulated tumor metabolism and accumulated metabolic waste.

VISTA is predominantly expressed on myeloid cells, including monocytes and neutrophils, where it acts as an immune checkpoint to suppress T-cell activation[3]. Importantly, VISTA is only active at low pH (~pH 6), as found in the TME, due to protonation of surface exposed histidine residues, enabling its binding to PSGL-1[7,8]. Despite the therapeutic potential of VISTA inhibition demonstrated in preclinical studies[2], clinical development of anti-VISTA antibodies has been challenging due to: 1) uncertainty about the critical counter-receptor responsible for T-cell suppression; 2) high drug clearance via target-

[1]Sensei Biotherapeutics Inc., 1405 Research Blvd, Suite 125, Rockville, MD 20850, USA. [2]genOway, Technopark Gerland, 69007 Lyon, France. [3]Department of Pathology and Immunology, Washington Univ. School of Medicine, Mailstop 8118, 425 South Euclid Ave, St. Louis, MO 63110, USA. ✉e-mail: evanderhorst@senseibio.com

mediated drug disposition (TMDD) by VISTA$^+$ neutrophils and monocytes at physiologic pH, and; 3) cellular activation and cytokine release syndrome (CRS) at sub-therapeutic doses due to engagement of VISTA in the blood, reducing the likelihood of reaching efficacious target occupancy levels in tumors. For example, among several non-pH-selective antibodies in clinical development, JNJ-61610588 (now CI-8993) induced dose-limiting on-target CRS at sub-therapeutic dose levels and exhibited TMDD[9].

We developed SNS-101, a fully human monoclonal IgG1 antibody (mAb) that is cross-reactive to cynomolgus monkey VISTA but not to mouse VISTA. SNS-101 is specific for the protonated (active) form of VISTA. We characterized SNS-101's pH-selective binding profile and determined its epitope on VISTA. Furthermore, we compared SNS-101 to several clinical stage non-pH-selective anti-VISTA antibodies in in vitro and in vivo pharmacology, PK, and safety studies in mice and non-human primates (NHP).

In addressing the challenges of VISTA-targeted cancer immunotherapy, we introduce SNS-101, a pH-selective antibody designed to mitigate rapid clearance and cytokine release syndrome, enhancing the therapeutic efficacy of PD-1 inhibitors. This study elucidates SNS-101's mechanism of action and its potential to shift the tumor microenvironment towards an anti-tumor state, setting the stage for its evaluation in ongoing clinical trials.

## Results

### A pH-selective antagonistic VISTA antibody

SNS-101 is a fully human IgG1 kappa mAb, which was discovered from a yeast-based library comprising highly diverse synthetic human immune repertoires[10]. The library was subjected to iterative rounds of positive enrichment and negative selection cycles at pH 6.0 (Supplementary Fig. 1). Given the reported TME range of pH 5.6-6.8[11,12], we characterized SNS-101's interaction with human VISTA across a pH range of 5.8–7.4. We observed a continuous decrease in binding affinity from 0.35 nM to 353 nM as the pH increased (Fig. 1a-b, Supplementary Table 1), corresponding to a decrease and increase in the association and dissociation rate constant respectively. Importantly, high affinity binding ($K_D = 0.35$-5.9 nM) was observed at pH levels up to 6.4, indicating SNS-101's potential effectiveness in the acidic TME.

### Histidines and arginines in a unique VISTA epitope contribute to SNS-101's pH-selectivity

SNS-101's highly pH-dependent binding to VISTA suggests the involvement of histidine residues in this interaction. The extracellular domain (ECD) of VISTA, an IgV-type domain, has a significantly higher histidine content compared to other B7 family and type I transmembrane proteins (14 surface exposed histidines out of total 168 residues)[13]. To elucidate the exact nature of SNS-101's pH-selective binding, we co-crystallized the VISTA-ECD with a SNS-101 Fab fragment (Fig. 1c and Supplementary Table 2). The structure of VISTA:SNS-101 Fab at 2.59 Å resolution revealed that five out of fourteen histidines of VISTA-ECD (His98, His100, His117, His154 and His155) are located at the interface of this complex (Supplementary Fig. 2). In addition to histidines, we identified all VISTA residues within a 4 Å distance to the SNS-101 surface with potential side chain interactions to the heavy or light chain of SNS-101. These potentially critical VISTA residues were mutated to alanine, and their influence on the binding affinity was quantified by surface plasmon resonance (SPR) (Supplementary Figs. 3 and 4, Supplementary Table 3). Among all interface histidine residues, only single histidine mutations His98 ($K_D = 9.5$ nM), His100 ($K_D = 7.5$ nM) and His117 ($K_D = 4.7$ nM) significantly decreased binding affinity by 14-, 11- and 7-fold, respectively (Supplementary Figs. 3 and 4, Supplementary Table 3). Double (His98Ala/His100Ala; $K_D = 23$ nM) and triple (His153Ala/His154Ala/His155Ala; $K_D = 17$ nM) histidine mutant variants decreased binding affinity by 33- and 25-fold. Combining His98Ala/His100Ala with His154Ala/His155Ala or His117Ala mutations

further reduced the binding affinity 240-fold ($K_D = 166$ nM) and 130-fold ($K_D = 90$ nM), respectively (Supplementary Figs. 3 and 4, Supplementary Table 3).

Among the non-histidine interface residues, Arg159, Phe94 and Arg86 mutants significantly impacted binding affinity. Single mutant variants Arg159Ala and Phe94Ala reduced the binding affinity 180-fold ($K_D = 122$ nM) and 86-fold ($K_D = 59$ nM) respectively, while Arg86Ala entirely eliminated the interaction with SNS-101.

Arginine 159, positioned within 3 Å of Asp92 of the SNS-101 light chain (SNS-101-Fab:LC), is crucial for the high-affinity interaction. Mutating Asp92 nearly eradicated the interaction between SNS-101 and VISTA, indicating the importance of hydrogen bonds (H-bonds) between Arg159:VISTA and Asp92:SNS-101-Fab:LC (Fig. 1d). The diminished binding affinity in Arg86Ala and Phe94Ala variants can be explained by H-bonds between Arg86:VISTA and the backbone carbonyl oxygen of Asp92/Ala91 of SNS-101-Fab:LC, and -π-π-interactions between Phe94:VISTA and Phe93 of SNS-101-Fab:LC (Fig. 1d). Our data demonstrate that mutating His98/His100/His154/His155/Arg159 or His98/His100/His117/Arg159 either resulted in no interaction or 700-fold lower binding affinity to SNS-101 respectively, displaying the critical role of these residues in the pH-selective interaction between SNS-101 and VISTA (Supplementary Fig. 4 and Supplementary Table 3). In summary, the biochemical data support the crystal structure analysis, and highlight the importance of histidine residues in the identified pH-selective epitope of SNS-101.

### Comparison of SNS-101 epitope with other known VISTA antibodies

Next, we compared the SNS-101 epitope to VISTA antibodies from published literature and patents: HMBD-002[14], JNJ (VSTB174, Patent WO2017175058), SG7[15], 150.1 and 474.1 (Patent WO2022178203) and BMS767[7] (Fig. 1e-j). HMBD-002 binds in the proximity of the C-C' loop and β-sheets which includes residues 69–97 of VISTA-ECD (Fig. 1f). The binding epitope of JNJ (Arg86, Tyr69, Phe94, Gln95, His154) is mostly located within HMBD-002 epitope (Fig. 1g). SG7 binds Glu157, His154, Phe68, Lys70 close to the histidine-rich loop (residues 151-160) and barely interferes with binding epitopes of JNJ or HMBD-002 (Fig. 1h). The epitope of 150.1/474.1 (Tyr69, Arg86, Val149, Arg159) shares a small part of both histidine rich loop and C-C' β-sheet (Fig. 1i). Among the VISTA antibodies tested, only BMS767 is pH-selective and, similar to SNS-101, interacts with a larger portion of both histidine rich loop and C-C' β-sheet (Fig. 1j). Nevertheless, SNS-101's epitope (Arg159, His155, His154, His117, His100, His98, Arg86, Phe94) is distinct, involves more histidine residues and covers a larger VISTA-ECD surface area than BMS767 (His153, His154, Glu157, Val149, Leu147, Phe94, Gln95, Arg86, Thr71, Tyr69) (Fig. 1e). Taken together, our data identify and map a distinct epitope for SNS-101 compared to other VISTA antibodies.

### SNS-101 selectively inhibits VISTA interaction with potential partners at specific pH levels

Several partners have been described as VISTA receptors or ligands. We tested the interaction between VISTA and PSGL-1[7], V-set and Ig domain-containing 3 (VSIG-3)[16], VSIG-8[17], Syndecan-2[18] and leucine-rich repeat Ig-like domains (LRIG-1; WO2019165233, WO2021047104) at pH 6.0 and pH 7.4 (Fig. 2a). Only PSGL-1 and Syndecan-2 displayed significant dose-dependent interactions at pH 6.0, with respective binding enhancements of 125-fold and 78-fold compared to their minimal interactions at pH 7.4 (Fig. 2a). The pH-selective interaction between VISTA and PSGL-1 was also confirmed by SPR (Supplementary Fig. 5). Competition experiments at pH 6.0 showed that SNS-101 inhibited the interaction between VISTA and all tested binding partners with a 50% of the maximal inhibitory concentration ($IC_{50}$) of 5.7 to 6.7 nM (Fig. 2b-f). We also performed competition experiments with the non-pH-sensitive VISTA antibodies JNJ and h26A (Fig. 2g-k). Our observations indicate that JNJ was particularly effective in

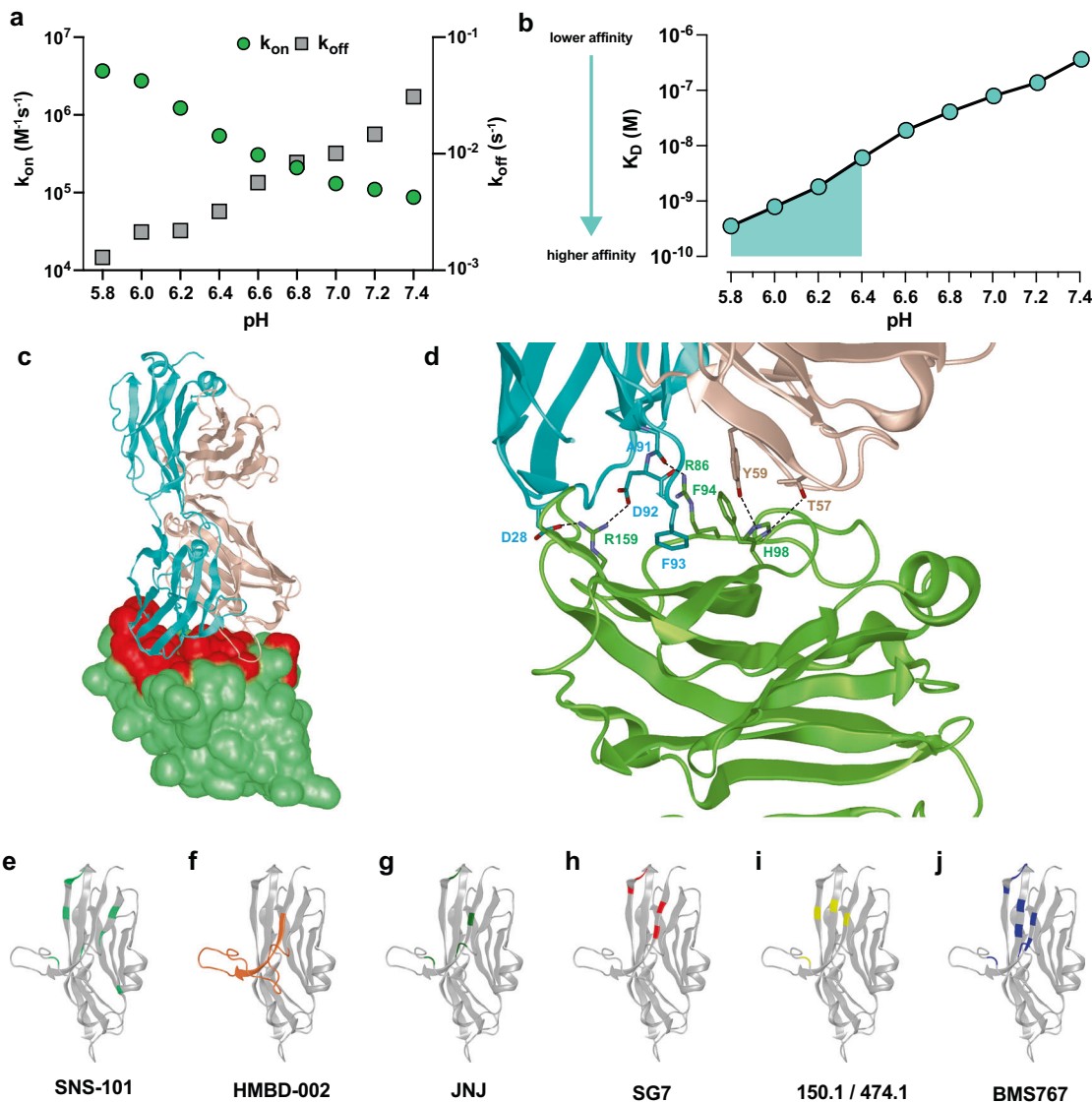

**Fig. 1 | Biophysical and structural characterization of interaction between SNS-101 and VISTA.** The pH-dependent (**a**), association ($k_{on}$, green) and dissociation ($k_{off}$, gray) rate constants and (**b**), equilibrium dissociation ($K_D$, cyan) constants for the VISTA:SNS-101 interaction were determined by SPR. The highlighted pH range of 5.8 to 6.4 (cyan) corresponds to single-digit nM $K_D$ measurements. **c** The crystal structure of the VISTA:SNS-101-Fab complex is shown with VISTA and SNS-101-Fab depicted in molecular surface (green) and ribbon cartoon (cyan−LC; Irish cream−HC), respectively. The surface of VISTA blocked by SNS-101 (including all VISTA residues within 4 Å of SNS-101) is marked in red. **d** Interface of the VISTA:SNS-101-Fab complex and the interaction between critical residues essential for high-affinity binding between VISTA and SNS-101. **e−j** Contrasting the VISTA epitopes targeted by SNS-101 and other anti-VISTA antibodies. Epitope residues important for cognate interactions between antibody and VISTA are shaded in green (SNS-101), orange (HMBD-002), dark green (JNJ), red (SG7), yellow (150.1 and 474.1; Kineta) and blue (BMS767).

inhibiting the interaction between VISTA and VSIG-3 ($IC_{50} = 4.5$ nM), supporting previous results[19], and Syndecan-2 ($IC_{50} = 4.7$ nM). In contrast, h26A exhibited the highest potency in inhibiting the interaction between VISTA and Syndecan-2 ($IC_{50} = 1.9$ nM) and LRIG-1 ($IC_{50} = 2.0$ nM). These findings are consistent with the identification of a distinct epitope for SNS-101.

Further, we conducted expression analysis of PSGL-1, Syndecan-2, LRIG-1, VSIG-3 and VSIG-8 on primary T-cells by flow cytometry (Supplementary Fig. 6). The results reveal expression of only PSGL-1 and, to a lesser extent LRIG-1. Notably, Syndecan-2, VSIG-3 and VSIG-8 were not detected. The absence of VSIG-3 on T-cells aligns with a previous report[7]. Subsequently, we evaluated the inhibition of VISTA binding to its receptor, sulfated PSGL-1, by SNS-101 at pH 6.0. This was done using flow cytometry on activated human CD4+ and CD8+ T-cells, as these cells express sulfated PSGL-1 on their surface[20]. The results show that SNS-101 induced a dose-dependent inhibition of VISTA's interaction

with human CD4+ and CD8+ T-cells at pH 6.0 with 50% of the maximal effective concentration ($EC_{50}$) values of 73 nM and 49 nM, respectively (Fig. 2l, m and Supplementary Fig. 7).

## SNS-101 shows mitigated CRS risk in vitro and in vivo

Targeting VISTA with a non-pH-selective antibody led to dose-limiting CRS in patients, ending the clinical trial (NCT02671955)[9]. To address this critical issue, we assessed the CRS potential of SNS-101 in a variety of in vitro and in vivo studies. As activation of endothelial cells seems to be a hallmark of severe CRS[21], we first assessed cytokine release in co-cultures of human umbilical vein endothelial cells (HUVECs) and human PBMCs[22]. In this assay, SNS-101 was compared to two non-pH-selective VISTA mAbs, JNJ (Patent WO2017175058) and h26A (Patent WO2016094837A2) at pH 7.4 (Fig. 3). Non-pH-selective h26A resulted in the highest level of induction of all 8 cytokines tested (Fig. 3a−h; blue). JNJ elicited strong induction of IL-8, and mild induction of tumor

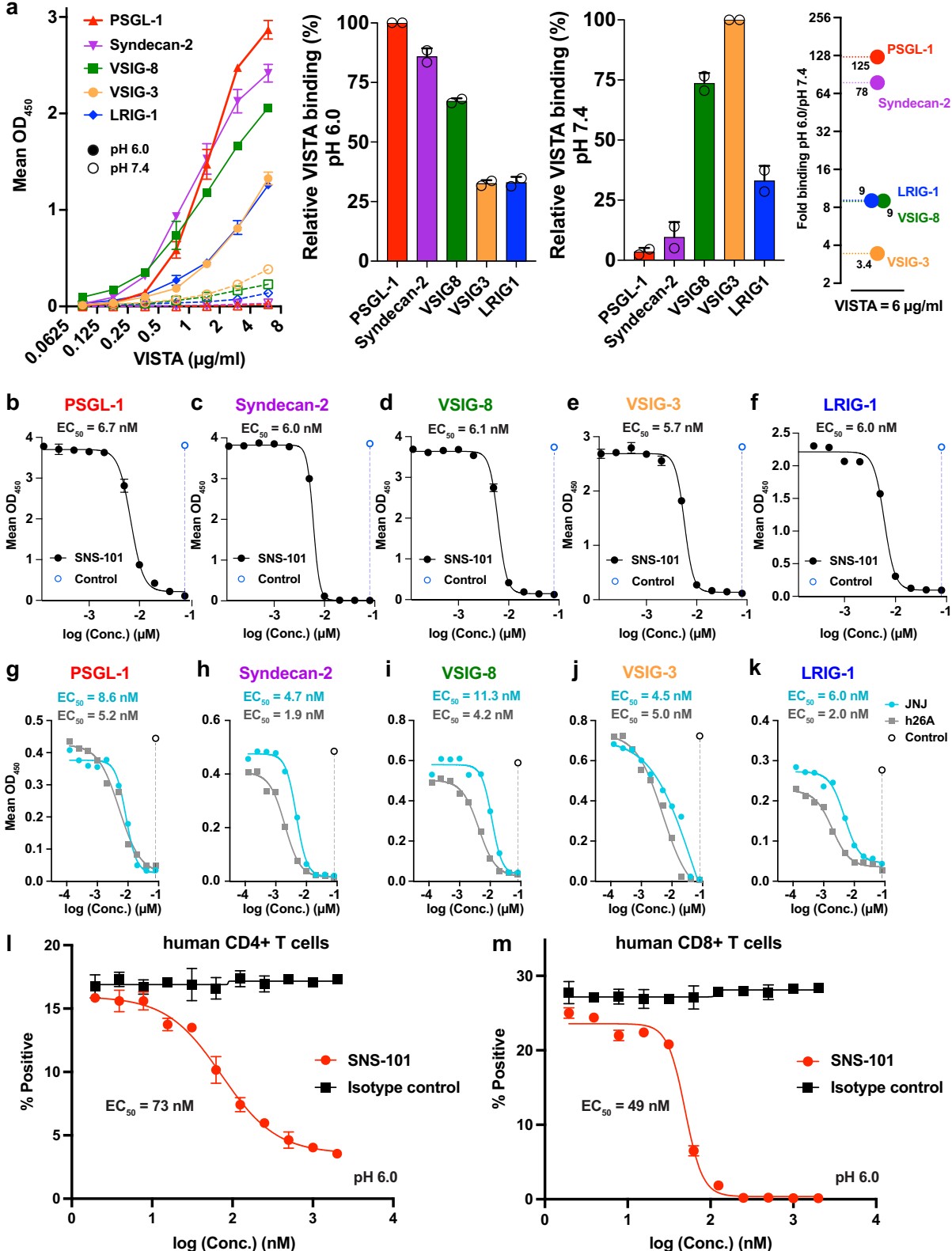

**Fig. 2 | Binding of VISTA to putative partners and competition by SNS-101.**
**a** ELISA experiment showing VISTA binding to immobilized PSGL-1 (red),
Syndecan-2 (purple), VSIG-8 (green), VSIG-3 (orange) and LRIG-1 (blue) at pH 6.0
(solid symbols) or pH 7.4 (open symbols). Middle panels show relative VISTA
binding to the most potent interaction, while the right panel indicates fold-
difference in binding at pH 6.0 or pH 7.4 at VISTA concentration of 6 μg/mL. Data
are presented as mean values ± SD. **b**–**f** ELISA assessing SNS-101's (black) disrup-
tion of the interaction between VISTA and binding partners from (**a**). Data are

presented as mean of technical replicates. **g**–**k** ELISA measuring disruption of the
interaction between VISTA and binding partners from (**a**) by JNJ (cyan) or h26A
(gray). Competition assay using PSGL-1[+] human CD4[+] T-cells (**l**), or CD8[+] T-cells (**m**) by
SNS-101 (red) or isotype control (black) at pH 6.0. Data are presented as mean
values ± SD. $EC_{50}$ values determined by curve fitting using GraphPad Prism are
indicated. $N = 2$ independent replicate experiments throughout. Source data are
provided as a Source Data file.

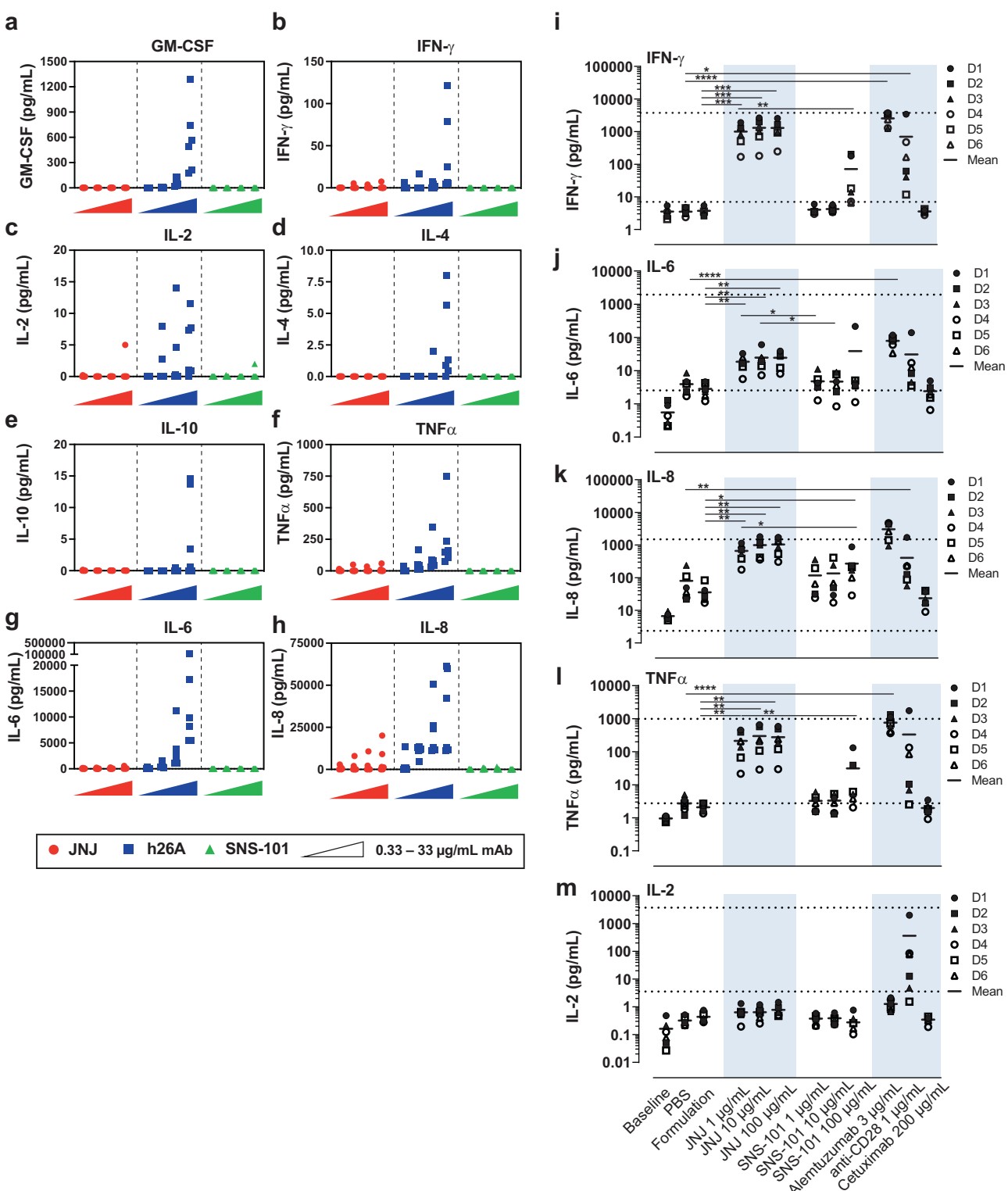

necrosis factor (TNF)-α and interferon (IFN)-γ (Fig. 3b, f, h; red). In contrast, SNS-101 did not induce significant secretion of any of the 8 screened cytokines at any doses, except for one donor releasing low levels of IL-2 at the highest dose (Fig. 3c; green).

We then employed a circulating human whole blood loop ex vivo assay for a more accurate CRS prediction[23]. SNS-101 and JNJ were tested at concentrations of 1, 10, and 100 mg/mL, mirroring anticipated clinical plasma levels. Subsequently, we assessed cytokine release and activation of platelets (PLT), neutrophils, monocytes, and NK cells (Fig. 3i–m, Supplementary Fig. 8).

In the ex vivo assay, JNJ significantly increased levels of IFN-γ, IL-6, IL-8, and TNF-α across all tested concentrations compared to SNS-101 (Fig. 3i–l). Even at lower concentrations (1 or 10 μg/mL), JNJ elevated IL-6 levels more than SNS-101 at 100 μg/mL (Fig. 3j). SNS-101 only significantly raised IL-8 levels at 100 μg/mL (Fig. 3k). Only JNJ significantly reduced PLTs and increased leukocyte-bound platelet activation at 100 μg/mL (Supplementary Fig. 8). Both JNJ (at all concentrations) and SNS-101 (only at 100 μg/mL) significantly activated monocytes and NK cells (Supplementary Fig. 8).

**Fig. 3 | In vitro and ex vivo CRS assessment; cytokine release. a–h** Cytokine release after incubation of soluble JNJ (red), h26A (blue) or SNS-101 (green) mAbs in HUVEC:PBMC co-culture assays. Eight cytokines in the culture supernatants from mAb-treated co-cultured HUVEC and PBMCs (from 6 different human donors) were quantified with 6 replicates for each concentration of each mAb. Each dot represents the result from one donor. Antibody concentrations of 0.33, 3.33, 10 and 33.3 μg/ml were tested. **i–m** Cytokine release in freshly collected, circulating blood from healthy human volunteers. JNJ and SNS-101 were tested at the concentrations indicated. Appropriate positive controls with known effects on the test parameters were included, i.e., alemtuzumab (anti-CD52), and anti-CD28. Phosphate buffered saline (PBS), cetuximab (anti-epidermal growth factor receptor [EGFR] antibody) and formulation buffer for SNS-101 and JNJ, were used as negative controls. A single n = 1 experiment was performed with n = 6 independent donor samples. Statistical analyses were performed with Paired Student's $t$ test on Log10 transformed values above LLOQ, with Holm-Sidak correction for multiple comparisons (*0.01 < $P$ < 0.05; ** 0.001 < $P$ < 0.01; *** 0.0001 < $P$ < 0.001; ****$P$ < 0.0001). Exact $P$ values were as follows (non-significant values ($P$ > 0.05) not reported): IFN-γ: PBS vs. Anti-CD28, 1 μg/ml, $P$ = 0.0145; PBS vs. Alemtuzumab, 3 μg/ml, $P$ = 0.00000174; Formulation Buffer (FB) vs. JNJ 1 μg/ml, $P$ = 0.000137; FB vs. JNJ 10 μg/ml, $P$ = 0.000137; FB vs. JNJ 100 μg/ml, $P$ = 0.0000873; JNJ, 1 μg/ml vs. SNS-101, 1 μg/ml, $P$ = 0.000320; JNJ, 10 μg/ml vs. SNS-101, 10 μg/ml, P = 0.000320; JNJ, 100 μg/ml vs. SNS-101, 100 μg/ml, $P$ = 0.00142. IL-6: PBS vs. Alemtuzumab, 3 μg/ml, $P$ = 0.0000758; FB vs. JNJ 1 μg/ml, P = 0.00431; FB vs. JNJ 10 μg/ml, $P$ = 0.00431; FB vs. JNJ 100 μg/ml, $P$ = 0.00431; JNJ, 1 μg/ml vs. SNS-101, 1 μg/ml, $P$ = 0.0129; JNJ, 10 μg/ml vs. SNS-101, 10 μg/ml, $P$ = 0.0129. IL-8: PBS vs. Alemtuzumab, 3 μg/ml, $P$ = 0.00238; FB vs. JNJ 1 μg/ml, $P$ = 0.00136; FB vs. JNJ 10 μg/ml, $P$ = 0.00129; FB vs. JNJ 100 μg/ml, $P$ = 0.00116; FB vs. SNS-101, 100 μg/ml, $P$ = 0.0186; JNJ, 1 μg/ml vs. SNS-101, 1 μg/ml, $P$ = 0.0429; JNJ, 10 μg/ml vs. SNS-101, 10 μg/ml, $P$ = 0.0429; JNJ, 100 μg/ml vs. SNS-101, 100 μg/ml, $P$ = 0.0374. TNFα: PBS vs. Alemtuzumab, 3 μg/ml, $P$ = 0.00000315; FB vs. JNJ 1 μg/ml, $P$ = 0.00160; FB vs. JNJ 10 μg/ml, $P$ = 0.00137; FB vs. JNJ 100 μg/ml, $P$ = 0.00124; JNJ, 1 μg/ml vs. SNS-101, 1 μg/ml, $P$ = 0.00282; JNJ, 10 μg/ml vs. SNS-101, 10 μg/ml, $P$ = 0.00282; JNJ, 100 μg/ml vs. SNS-101, 100 μg/ml, $P$ = 0.00282. Data are presented as individual values with line at mean. Source data are provided as a Source Data file.

In summary, JNJ or h26A showed a stronger pro-inflammatory profile in vitro, while SNS-101 only induced mild effects at the highest concentration.

Previous studies demonstrated that the human CD34[+] cell-reconstituted BRGSF-HIS mouse model can functionally recapitulate human lymphoid and myeloid compartments[24]. Expanding on these insights, we explored SNS-101's potential for CRS with the same in vivo system (Fig. 4, Supplementary Fig. 9).

In vivo, anti-CD3 (OKT3, positive control) predictably induced inflammatory cytokine release. A trend was observed in monocyte activation (Supplementary Fig. 10) and CD8[+] T-cell proportions, though neither reached statistical significance. Notably, as expected, CD4[+] T-cell proportions were significantly reduced[25] (Fig. 4a–k, purple). Conversely, SNS-101, except for minimal C-C motif chemokine ligand (CCL)-5 release, did not trigger cytokine secretion at any dose (Fig. 4e, orange, red), and had no significant impact on CD14[+] monocyte activation, depletion, or T-cell proportions (Fig. 4k–l, Supplementary Fig. 10).

## SNS-101 exhibits a favorable PK and safety profile

The abundance of VISTA[+] cell populations in the periphery poses a significant pharmacokinetic (PK) challenge for non-pH-selective VISTA biologics. As SNS-101 does not cross-react with mouse VISTA, the PK profile of SNS-101 was assessed in hVISTA-KI mice with (hVISTA-KI-T) or without (hVISTA-KI-NT) tumor burden (MB49) as well as in wildtype (WT) C57BL6/J mice by single intravenous injection of 5 mg/kg (Fig. 5a). Serum clearance was slowest in the C57BL6/J mice (0.6 mL/h/kg) and similar to hVISTA-KI-NT mice (0.9 mL/h/kg), but fastest in the hVISTA-KI-T mice (1.3 mL/h/kg). Terminal half-life ($T_{1/2}$) and blood mean residence time (MRT) were similar in the hVISTA-KI-NT (184 h, 154 h) and C57BL6/J groups (180 h, 176 h) and fastest in the hVISTA-KI-T (70 h, 108 h). The difference between the two VISTA groups illustrates the effect of the presence of tumors on the disposition of SNS-101.

Additionally, we assessed the PK profile of SNS-101 in NHPs and compared it to the non-pH-selective antibody h26A, both of which cross-react with cynomolgus VISTA (Fig. 5b). As expected, h26A displayed pronounced TMDD with serum concentrations below detectable levels by 48 hours. In contrast, SNS-101 displayed dose-proportional exposure at 1, 10, or 100 mg/kg of SNS-101 in NHPs, with $T_{1/2}$ of 21-23 days across all dose-levels. This resulted in >19,000-fold difference in dose-normalized drug exposure (AUC/D) between h26A (0.209 ± 0.009 h*kg*mg/mL/mg) and SNS-101 (4000 ± 404 h*kg*mg/mL/mg) at the same 10 mg/kg dose. Our findings in hVISTA-KI mice and NHPs show linear disposition characteristics of SNS-101, consistent with the absence of TMDD.

Finally, the toxicity of SNS-101 was evaluated in NHPs through repeat administrations of 3, 10 and 100 mg/kg biweekly for three doses, followed by a six-week recovery period. Comprehensive clinical endpoints, including immunophenotypic and cytokine analyses and anatomic pathology evaluations, were assessed (Supplementary Data File 1).

SNS-101 was well tolerated at all dose levels, with no observable changes in clinical observations, body weight, clinical chemistry, macroscopic or microscopic findings, or peripheral blood immunophenotyping. Additionally, no alterations in plasma IFN-γ, TNF-α, or IL-6 concentrations were attributed to SNS-101 at any time point or dose level, substantiating previous in vitro, ex vivo and in vivo CRS model data (Supplementary Data File 1).

## SNS-101 re-sensitizes anti-PD-1 insensitive tumors and shifts cytokine/chemokines balance to an inflammatory M1-like phenotype

While the anti-tumor activity of VISTA biologics has been previously described[2–7,14,26–29], we specifically assessed the anti-tumor activity of SNS-101, both as standalone treatment and in combination with anti-mouse PD-1 (anti-mPD-1) across various syngeneic mouse models (Fig. 6 and Supplementary Fig. 11).

In the MC38 tumor model, SNS-101-m2 monotherapy at doses of 10 and 30 mg/kg modestly inhibited tumor growth by 27%, and 26%, respectively, but only reached statistical significance at the highest dose ($P$ = 0.04; Fig. 6a, b). Using a low dose of anti-mPD-1 (1 mg/kg; 1/10 of standard dose), to mimic tumor resistance, we observed negligible tumor growth inhibition (5%, $P$ > 0.05). However, combination of 10 and 30 mg/kg SNS-101-m2 with 1 mg/kg of anti-mPD-1 resulted in statistically significant tumor growth inhibition (52%, $P$ < 0.01 and 53% $P$ < 0.02, respectively) compared to anti-mPD-1 group at 1 mg/kg (Fig. 6a, b).

In the MB49 bladder carcinoma model, a moderate dose of anti-mPD-1 (5 mg/kg) significantly inhibited tumor growth (55%, $P$ < 0.001; Fig. 6c, d). Interestingly, even a modest dose of SNS-101 (3 mg/kg) as monotherapy also reduced tumor burden by 46%, ($P$ < 0.001). Their combination further amplified this effect, substantially impairing tumor growth by 83% ($P$ < 0.0001; Fig. 6c, d).

In the MCA/1956[30] sarcoma model, (Fig. 6e, f), we employed a standard dose of anti-mPD-1 (10 mg/kg) and focused on tumor response (Fig. 6e) and overall survival (Fig. 6f) as primary endpoints. The combination of SNS-101 with anti-mPD-1 led to an increase in tumor rejection (5/8 CR) compared to anti-mPD-1 alone (1/8 CR) and significantly extended overall survival ($P$ < 0.05). Notably, SNS-101 monotherapy showed no efficacy in the EG.7 syngeneic model, where anti-CTLA-4 was highly efficacious (Supplementary Fig. 11).

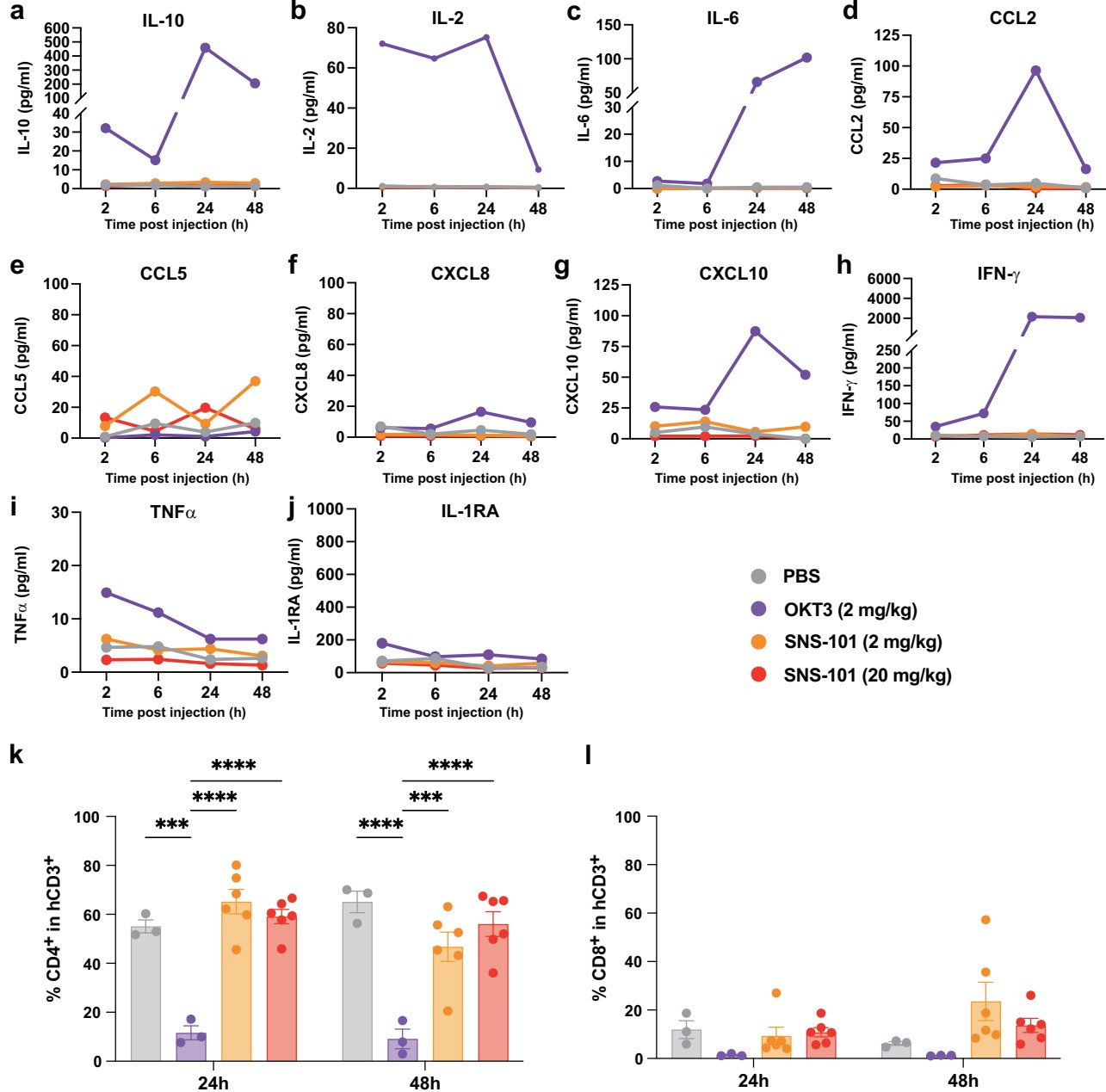

**Fig. 4 | In vivo CRS assessment in human CD34⁺ cell-reconstituted BRGSF-HIS mice. a–j** BRGSF-HIS mice (n = 6 per time point) were dosed with antibodies (OKT3 [2 mg/kg], purple; SNS-101 [2 mg/kg], orange; SNS-101 [20 mg/kg], red) or PBS (gray) by IV injection and blood was collected at 2-, 6-, 24- and 48-h post injection. Cytokines were quantified from serum by a Multiplex bead-based assay. Positive control anti-CD3 (OKT3) efficiently induced multiple inflammatory cytokines indicative of CRS (purple). **k–l** Flow cytometry analysis of splenocytes isolated from mice 24- and 48-h post-antibody injection. **k** Percentage of CD4⁺ T-cells in the CD45⁺CD3⁺ population. **l** Percentage of CD8⁺ T-cells in the CD45⁺CD3⁺ population. n = 3 biologically independent samples for PBS and OKT3 conditions; n = 6 for SNS-101 conditions. Colors indicate same conditions as above. Statistical analysis was performed using 2-way ANOVA Multiple comparison (* 0.01 < P < 0.05; ** 0.001 < P < 0.01; *** 0.0001 < P < 0.001; **** P < 0.0001). Exact P values were as follows: 24 h: PBS vs. OKT3 2 mg/kg, P = 0.0002; PBS vs. SNS-101 2 mg/kg, ns P = 0.5376; PBS vs. SNS-101 20 mg/kg, ns P = 0.9480; OKT3 2 mg/kg vs. SNS-101 2 mg/kg, P < 0.0001; OKT3 2 mg/kg vs. SNS-101 20 mg/kg, P < 0.0001; SNS-101 2 mg/kg vs. SNS-101 20 mg/kg, ns P = 0.7534. 48 h: PBS vs. OKT3 2 mg/kg, P < 0.0001; PBS vs. SNS-101 2 mg/kg, ns P = 0.0943; PBS vs. SNS-101 20 mg/kg, ns P = 0.6354; OKT3 2 mg/kg vs. SNS-101 2 mg/kg, P = 0.0002; OKT3 2 mg/kg vs. SNS-101 20 mg/kg, P < 0.0001; SNS-101 2 mg/kg vs. SNS-101 20 mg/kg, ns P = 0.4427. Data are presented as mean values ± SEM. Source data are provided as a Source Data file.

Collectively, our in vivo data suggest that VISTA inhibition by pH-selective SNS-101 augments anti-PD-1 responses in syngeneic mouse tumor models.

MC38 tumors were excised at the end of the study to enumerate tumor-infiltrating CD4⁺ and CD8⁺ T-cells (Fig. 7a). No changes in infiltrating CD4⁺ T-cells were detected in tumors (Supplementary Fig. 12). The groups treated with anti-mPD-1 (1 mg/kg) or SNS-101-m2 (10 mg/

kg, 30 mg/kg) exhibited 8.65%, 16.6% and 14.7% CD8⁺ T-cells within their tumors, respectively, which were not statistically significant compared to the 10.7% in the control arm (p > 0.05) (Fig. 7a). However, the combination of anti-mPD-1 with either 10 or 30 mg/kg of SNS-101-m2 significantly increased CD8⁺ T-cells to 19.3% and 26.4% (both P < 0.0001), respectively, correlating with the observed anti-tumor efficacy.

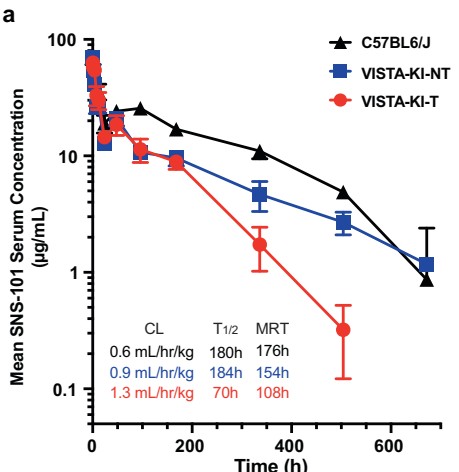

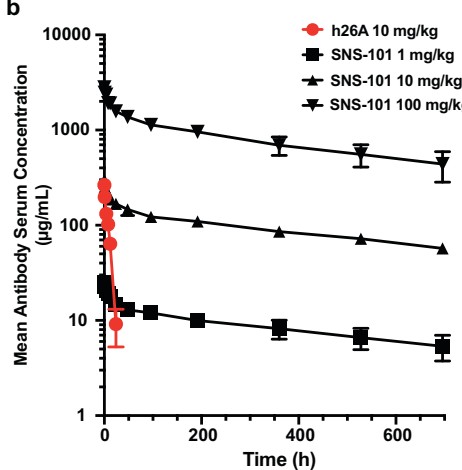

**Fig. 5 | PK profile of SNS-101 in mice and cynomolgus monkeys. a** WT C57BL6/J (black) and human VISTA knock-in (hVISTA-KI) mice with (red) or without (blue) MB49 tumors were compared. Tumors were established by subcutaneous injection ($1 \times 10^6$ cells; hVISTA-KI-T). Mice (n = 4 per group) were given a single intravenous injection of 5 mg/kg SNS-101. Serum samples were collected over a 28-day observation period and tested for the presence of SNS-101 by ELISA. Serum clearance (CL), terminal half-life ($T_{1/2}$) and blood mean residence time (MRT) are indicated. **b** Non-human primates (n = 4 per group) were administered a single intravenous infusion of SNS-101 (or comparator mAb h26A, red) over 1 h followed by a 28-day observation period. Circulating antibody was measured by ELISA. Data are presented as mean values ± SD. Source data are provided as a Source Data file.

We also collected plasma samples and profiled the levels of 49 murine cytokines and chemokines using multiplexed protein-based immunoassays. We first analyzed the results by hierarchical clustering and heatmap visualization (Fig. 7b). Correlation scores for the panel of cytokines/chemokines and tumor volume were used to identify the factors highly correlated and anti-correlated with tumor growth (Fig. 7c).

Investigating the relationship between cytokines and tumor volume, we established a strong linear correlation using a regression model with all cytokines as predicator variables, achieving a $R^2$ of 0.82. We then employed a machine learning approach, recursive feature elimination (RFE) by random forest classification, to identify the most relevant analytes from the multiplex panel. This analysis highlighted CCL-2, CCL-7, CCL-12, CCL-24, C-X-C motif chemokine ligand (CXCL)-10, CXCL-12 and IL-3 as key features (Fig. 7d). A reduced linear model incorporating only these seven cytokines captured most of the variation and effectively modeled tumor growth inhibition ($R^2 = 0.77$) upon treatment with SNS-101 and anti-PD-1 (Fig. 7d). Notably, treatment-induced tumor volume reduction correlated with decreased levels of CCL-2, CCL-7, and CCL-12 and increased levels of CXCL-10, CCL-24, CXCL-12, and IL-3.

## Discussion

The therapeutic potential of VISTA as an immune checkpoint has been challenging to explore due to the previously observed clinical manifestation of CRS[9]. To circumvent this, we developed SNS-101, a fully human antagonistic VISTA antibody that selectively binds to protonated VISTA at low pH, a critical feature to mitigate CRS.

Our structural and biochemical analyses demonstrate the significance of surface-accessible histidines and arginines in VISTA's ECD for SNS-101's pH-selective binding, resulting in an over 1000-fold decrease in binding affinity between pH 5.8 and 7.4. Most oxygen-rich tumor cells maintain an acidic pH of less than 6.5, and extracellular pH in tumors as low as 5.8 has been reported[31,32]. Moreover, the acidity is linked to tumor aggressiveness and is maintained even in metastatic cells surrounded by a pH 7.4 buffer[33,34]. The high binding affinity of SNS-101 in the pH range of 5.8 to 6.4 indicates that VISTA engagement can persist across low pH gradients in the TME.

Epitope comparisons to different VISTA antibodies currently in preclinical or clinical development reveal a unique epitope for SNS-101.

Importantly, this epitope distinguishes SNS-101 from a previously described pH-selective mAb[7].

Among the reported putative VISTA binding partners, PSGL-1 stands out due to its pH-dependent interaction with VISTA and its role as a negative checkpoint regulator on T-cells[35–37]. Johnston further corroborated this, observing that a knock-down of PSGL-1 in T-cells resulted in the loss of binding to recombinant VISTA at pH 6.0[7]. Our assessment indicates that only PSGL-1 and the monocyte proteoglycan Syndecan-2 bind to VISTA in a pH-dependent and significant manner. The critical histidines in VISTA (His154, His155, His98 and His100) for pH-selective binding with sulfated tyrosines in PSGL-1 (Tyr46, Tyr48, Tyr 51) are identical to residues identified in the SNS-101 epitope[7]. Critical amino acid residues for VSIG-3 (Arg86, Phe94, Gln95)[19] and LRIG-1 binding (Thr82, Arg87)[14] to VISTA either overlap with or are in close proximity to the SNS-101 epitope. This suggests that SNS-101's unique epitope encompasses VISTA ligand binding sites, overlaps with known epitopes of other anti-VISTA antibodies, and contains critical residues that introduce a strong pH selectivity. The role of Syndecan-2 as a novel regulator of VISTA binding to monocytic cells[18] and its pH-dependency remain to be investigated.

VISTA⁺ myeloid-lineage cells, including monocytes, neutrophils, and dendritic cells, contribute significantly to CRS development, particularly through the production of cytokines like IL-6[38,39]. Interestingly, agonistic targeting of VISTA has recently been proposed as a potential strategy to alleviate Chimeric Antigen Receptor (CAR) T-cell-induced CRS through myeloid compartment modulation[40]. Our human in vitro and ex vivo, murine in vivo CRS, and NHP safety data demonstrate that SNS-101 significantly mitigates, if not eliminates, cytokine release due its lack of VISTA binding at physiologic pH (Supplementary Fig. 13).

In addition to CRS, rapid clearance of VISTA antibodies presents a significant challenge in achieving clinically efficacious dose levels. The PK profile of SNS-101 in NHPs, in C57BL/6 mice, and hVISTA-KI mice without tumors displayed linear disposition characteristics and an absence of TMDD, as SNS-101 binds to VISTA only when VISTA becomes protonated. The typical parameter estimates were consistent with the PK of an IgG mAb in a non-target species, contrasting with other non-pH-selective VISTA antibodies[9,14,29,41]. Evaluating the PK profile of SNS-101 in VISTA-KI mice with or without tumor burden, as well as in C57BL6/J mice, showed that the presence of tumors affected the

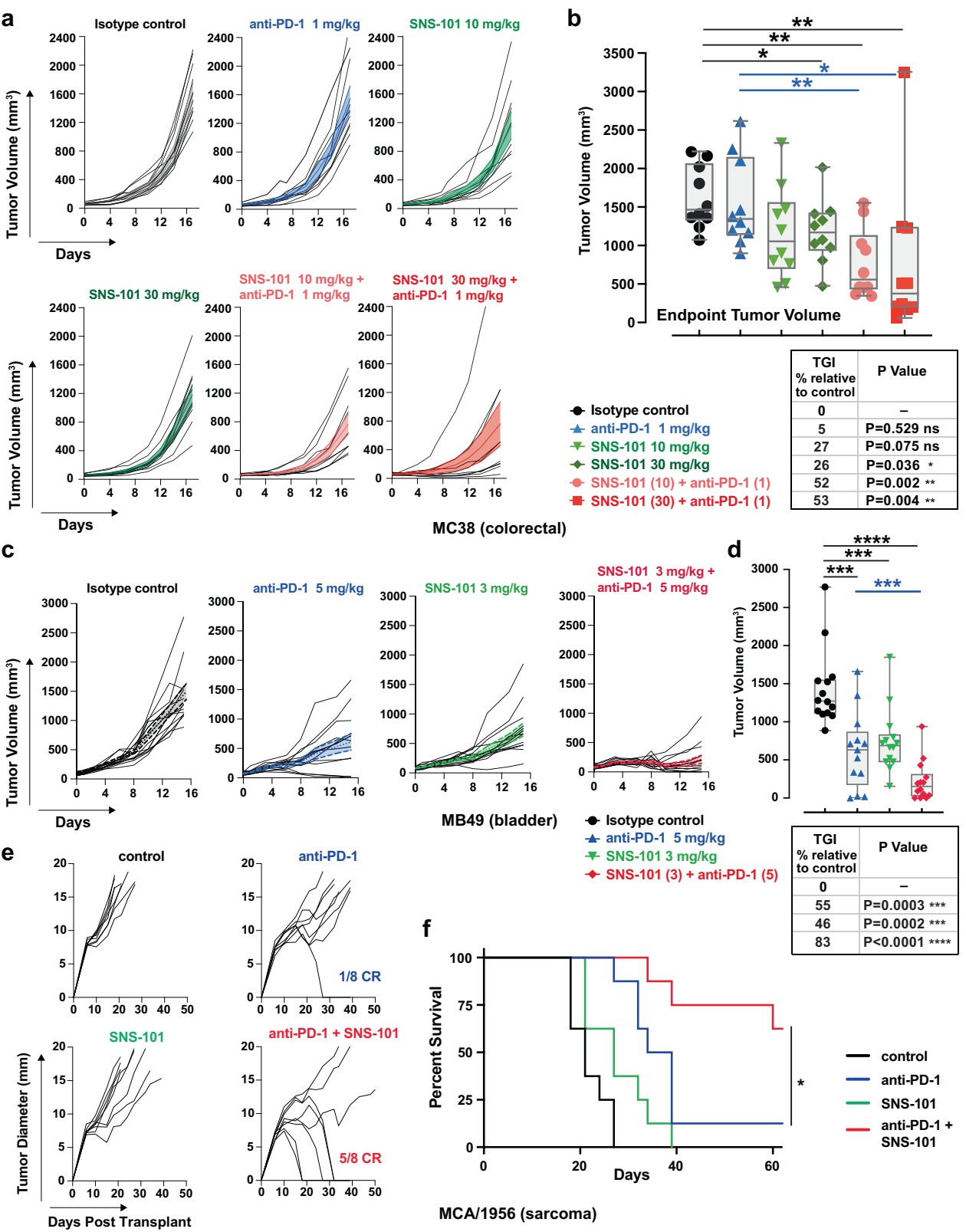

disposition of SNS-101. This is consistent with the proposed mechanism of action and biodistribution of SNS-101, as increased tumor volume would result in a higher amount of VISTA positive tumor-infiltrating immune-suppressive myeloid cells and an acidic TME.

In our study, we demonstrated the effectiveness of SNS-101 in combination with a murine PD-1 immune checkpoint inhibitor (ICI) using the MC38 colon, MB49 bladder and MCA/1956 sarcoma syngeneic tumor models. Our analysis of circulating cytokines and chemokines from MC38 tumors revealed widespread changes upon treatment with SNS-101 and anti-PD-1. Notably, our computational analysis of this high-dimensional data suggests that the key changes correlating with inhibition of tumor growth include the down-regulation of CCL-2, CCL-7, CCL-12, and the upregulation of CXCL-10, CCL-24, CXCL-12 and IL-3.

**Fig. 6 | In vivo efficacy of SNS-101 and anti-PD-1 combination in syngeneic mouse models demonstrates enhanced tumor growth inhibition.** Spider plots of tumor volume measurements for each mouse ((**a**) MC38; n = 12/group, (**c**) MB49; n = 14/group, (**e**) MCA/1956; n = 8/group) following subcutaneous injection of $1 \times 10^6$ cells (MC38, MB49) or $1.5 \times 10^6$ cells (MCA/1956) into hVISTA-KI mice (female MC38; 1956, male MB49). Color overlays represent means ± SEM. Seven days (MC38, MB49) or 10 days (MCA/1956) after implantation, mice were randomized and treated with the indicated doses of isotype control IgGs, SNS-101-m2 (variable regions of SNS-101 grafted onto a mouse IgG2a framework to reduce potential neutralizing antibody generation), anti-mPD-1 (clone RMP1-14, Bio X Cell) or combinations. Treatment regimen: **a, c** SNS-101-m2 was administered 3x per week and anti-mPD-1 2x per week for the duration of the experiment (3 weeks MC38; 2 weeks MB49). **e** Mice received isotype control IgG, a standard dose of anti-mPD-1 (10 mg/kg), SNS-101 (20 mg/kg) or the combination 2x per week. **b** Endpoint tumor volumes of MC38 tumors with statistical significance, evaluated using the Mann-Whitney two-sided unpaired t test (*$0.01 < P < 0.05$; ** $0.001 < P < 0.01$; *** $0.0001 < P < 0.001$; **** $P < 0.0001$). Exact $P$ values were as follows: Control vs. SNS-101 10, ns P = 0.0753; Control vs. SNS-101 30, P = 0.0355; Control vs. anti-PD-1 1/SNS-101 10, P = 0.0021; Control vs. anti-PD-1 1/SNS-101 30, P = 0.0039; anti-PD-1 1 vs. anti-PD-1 1/SNS-101 10, P = 0.0058; anti-PD-1 1 vs. anti-PD-1 1/SNS-101 30, P = 0.0147. Box plots display the 25th to 75th percentiles with a median line. Whiskers represent the min-max range from smallest value to largest value. n = 10 biologically independent samples (animals/group); **d** Endpoint tumor volumes of MB49 tumors with statistical significance, evaluated using the Mann-Whitney two-sided unpaired t test: n = 14 biologically independent samples (animals/group). Inset tables for (**b**) and (**d**) show tumor growth inhibition (TGI), calculated as described in Methods and corresponding statistical analysis (above). **f** Survival curve for the MCA/1956 sarcoma model, with significance determined using the Mantel-Cox log-rank test (*P = 0.02). CR = complete response. Tumor growth inhibition calculations and significance criteria are as described[60]. Source data are provided as a Source Data file.

These data support the notion that SNS-101 and PD-1 combination therapy induces a phenotypic shift in macrophages from M2 to M1[42,43]. Macrophage polarization occurs as a spectrum of phenotypes, with M1 and M2 representing opposite ends with divergent functions. M1-like macrophages are proinflammatory and generally display anti-tumor activity. These M1-like macrophages typically secrete TNF-α, IL-6, IL-12, IL-23 and CXCL9, CXCL10 and CXCL11. In contrast, M2-like macrophages are characterized by the secretion of TGF-β, CCL22, CCL24 and IL-10, amongst others[44,45]. Our cytokine profiling shows that many of these characteristic inflammatory M1-like cytokines and chemokines are increased upon treatment with SNS-101 and anti-PD-1, while levels of several M2-like immunomodulators decrease. Tumor-resident mesenchymal stem cells, known for their significant secretion of a variety of chemokines, including CCL-2, CCL-7 and CCL-12, enhance the recruitment of CCR2-expressing monocytes to tumor sites, thereby increasing the macrophage population. This not only amplifies tumor growth but also instigates a phenotypic shift in macrophages. Thus, observed decreases in CCL2, CCL7 and CCL12 suggest a reduction in factors that promote pro-tumor, M2-like macrophage differentiation and infiltration[46].

Furthermore, the combination of CCR2 (the receptor for CCL2) inhibition and anti-PD-1 therapy has been shown to enhance tumor responses over anti-PD-1 monotherapy through enhanced recruitment and activation of CD8+ T-cells[47]. Our findings of increased intratumoral CD8+ T-cells similarly suggest the induction of a more permissive immune microenvironment. We propose that the combination of SNS-101 and PD-1 blockade may indirectly impede the immune-suppressive M2-like macrophage network through distinct modulation of specific cytokine/chemokines.

While many patients experience therapeutic benefits from approved immune checkpoint inhibitors, primary and adaptive resistance challenges persist, limiting overall treatment efficacy. A key determinant of the ICI therapeutic response is the tumor immune microenvironment, often categorized as "hot" or "cold". The former indicates a high degree of cytotoxic T-cell infiltration (T-cell-inflamed) and increased sensitivity to ICIs[48,49]. To evaluate the impact of a biologic like SNS-101 in these distinct settings, we employed several models. The MC38 model received a low dose of anti-PD-1 to simulate resistance conditions. The MCA/1956 model is immunogenic and responsive to anti-PD-1 treatment at early timepoints but acquires resistance as tumors progress. Both MC38 and MCA/1956 served as representations of 'hot' tumors, whereas the MB49 model was used as proxy of 'cold' tumors in mice.

The observed distinct responses to anti-PD-1 and SNS-101 in these models underscore the nuanced therapeutic potential of SNS-101 in cancer immunotherapy, emphasizing the importance of context in evaluating its efficacy.

PSGL-1 has been identified as a negative checkpoint of CD4+ T-cells[35], and the role of CD4+ T-cells as a key contributor to immunotherapy efficacy and tumor immunity was recently reported[50–52]. Additionally, paracortical zones of lymph nodes, which are notably acidic, are enriched in CD4+ T-cells[53]. This acidic environment within the immune system could act as a self-regulating feedback mechanism where the VISTA:PSGL-1 interaction may play a key role in suppressing the response of effector T-cells.

VISTA is a significant and emerging target in the immuno-oncology field. Our data provide preclinical proof of concept for pH-sensitive targeting of VISTA with SNS-101. We propose a mechanism of action that supports phase I clinical evaluation of SNS-101 in solid tumors. SNS-101 has recently entered clinical trials as monotherapy and in combination with a PD-1 inhibitor in patients with advanced solid tumors (NCT05864144).

## Methods

### Ethics statement

All animal experiments were performed in specific-pathogen free facilities in accordance with American Association for Laboratory Animal Science guidelines. Non-human primate studies were approved by the Charles River Labs Institutional Animal Care and Use Committee (IACUC) committee. Mouse experiments were conducted at Murigenics and Washington University in St. Louis. All experiments with mice were approved by the Institutional Animal Care and Use Committee of Murigenics (IACUC) or the Washington University Animal Studies Committee (School of Medicine, Washington University in St. Louis).

### Protein preparation

For antibodies generated by transient transfection, synthetic genes encoding HC and LC sequences with appropriate restriction sites (GeneArt, Thermo Fisher Scientific) were cloned into expression vectors pFUSEss-CHIg-hG1 and pFUSE2ss-CLIg-hk, respectively (InvivoGen pfusess-hchg1 and pfuse2ss-hclk). The sequence of SNS-101 was disclosed in US20230272082A1. SNS-101-m2 was designed by grafting the variable regions of SNS-101 onto a mouse IgG2a framework. JNJ and h26A were obtained by grafting the respective variable regions of either JNJ-61610588 (WO2017175058) or 26A (WO2017175058) onto a human IgG1 framework. Heavy and light chains were co-expressed using the ExpiCHO Expression System (Thermo Fisher Scientific A29133) and antibody purification performed on a Protein A affinity column (HiTrap MabSelect SuRe resin (Cytiva 11003494)) with 0.1 M Sodium Citrate, pH 3.0 elution. After neutralization and pooling of main peak fractions, material was buffer exchanged into PBS (Lonza 17-517Q) by dialysis. An IEX polishing step was conducted as needed. Antibody material evaluated in the multi-dose tox study of SNS-101 in Cynomolgus monkeys was produced in a fed-batch bioreactor from a

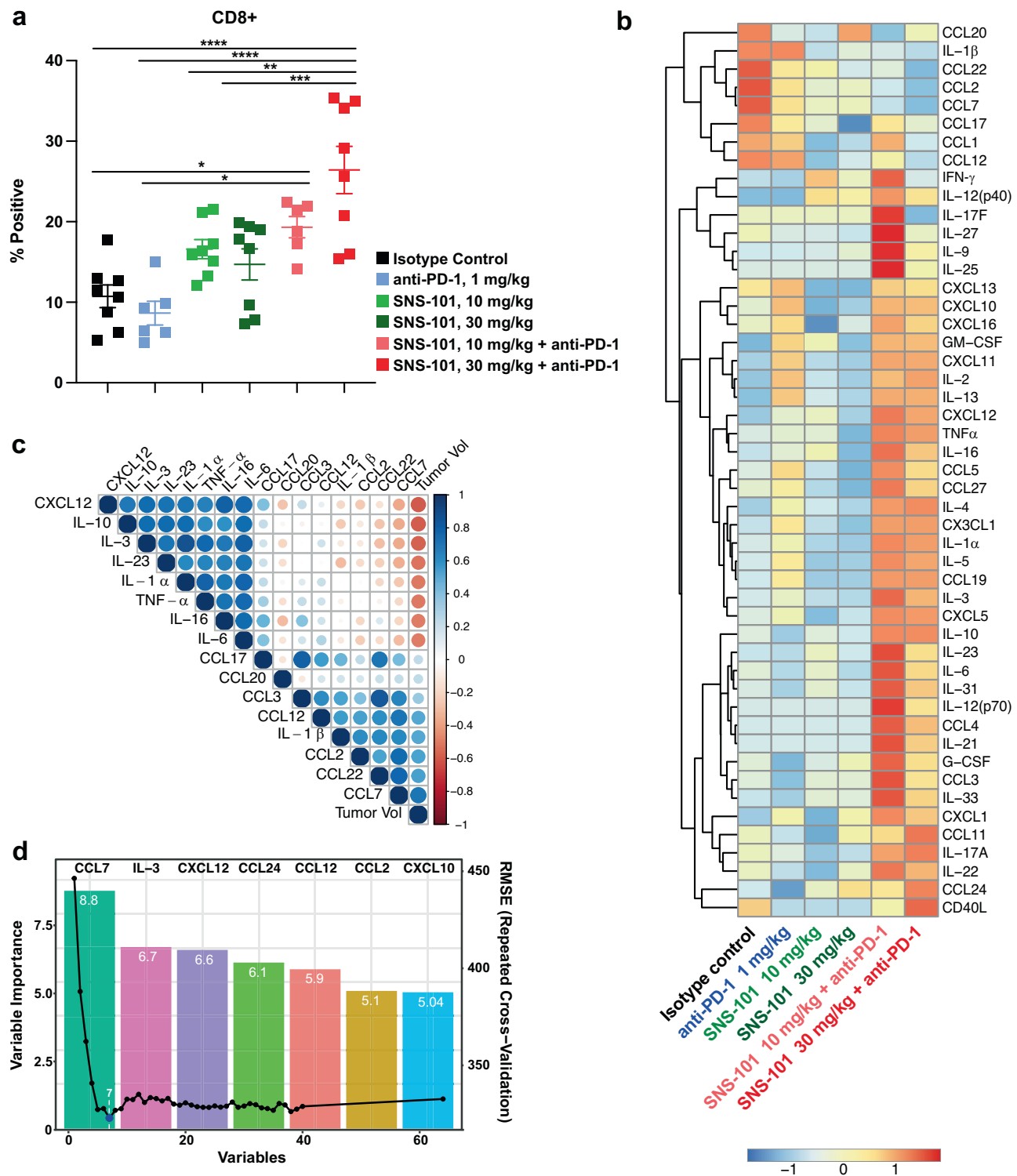

clonal CHO stable cell line. Clarification by depth filtration was followed by low pH hold viral inactivation, and a three-step chromatography process (Protein A capture, Anion and Cation exchange chromatography). Concentration and formulation were done using tangential flow filtration (TFF) (Catalent, Madison, WI).

VISTA-His, VISTA-His variants, VISTA without tag (for crystallography) and PSGL-1-19-mer-Fc were expressed using Expi293 expression system (Thermo Fisher Scientific A14635). Expression cloning vector pcDNA3.4-TOPO (Thermo Fisher Scientific) and pFUSEss-CHIg-hG1 (InvivoGen) were used for VISTA and PSGL-1-19mer-Fc respectively. The plasmid DNA for VISTA-His variants was either

obtained directly from GeneArt (Thermo Fisher Scientific) or prepared in-house using site-directed mutagenesis. VISTA-His/variants and untagged VISTA were purified using Ni-NTA resin (HisTrap™ Excel; Cytiva 17371206) and the purified protein was stored in PBS, pH 7.4 buffer. For crystallography studies, untagged VISTA, purified using Ni-NTA, was further polished with anion exchange chromatography (HiTrap™ Capto™ Q ImpRes; Cytiva 17547055). PSGL-1-19mer-Fc was co-expressed with plasmids encoding glucosaminyl (N-acetyl) transferase (core 2, hGCNT1; Origene RC214131) and alpha-(1,3)-fucosyl-transferase-7 enzymes (hFut3; Origene SC300017) at 2:1:1 ratio to enhance the post-translational modification (sialyl-Lewis X). The

**Fig. 7 | In vivo efficacy of SNS-101 and anti-PD-1 correlates with CD8$^+$ T cell infiltration and changes in key cytokine and chemokine networks.**
**a** MC38 tumors (Fig. 6) were excised, dissociated and proportions of CD8$^+$ T-cells among CD45$^+$ cells were measured by flow cytometry (n = 8 isotype control; n = 6 anti-PD-1; n = 8 SNS-101, 10 mg/kg; n = 8 SNS-101, 30 mg/kg; n = 6 SNS-101, 10 mg/kg + anti-PD-1, 1 mg/kg; n = 8 SNS-101, 30 mg/kg + anti-PD-1, 1 mg/kg). One-way ANOVA, with Tukey Post-Hoc testing was used for statistical analysis (*$0.01 < P < 0.05$; **$0.001 < P < 0.01$; ***$0.0001 < P < 0.001$; ****$P < 0.0001$). Exact $P$ values were as follows (non-significant values ($P > 0.05$) not reported): Control vs. SNS-101 10 mg/kg + anti-PD-1 1 mg/kg, $P = 0.0397$; Control vs. SNS-101 30 mg/kg + anti-PD-1 1 mg/kg, $P < 0.0001$; anti-PD-1 1 mg/kg vs. SNS-101 10 mg/kg + anti-PD-1 1 mg/kg, $P = 0.0108$; anti-PD-1 1 mg/kg vs. SNS-101 30 mg/kg + anti-PD-1 1 mg/kg, $P < 0.0001$; SNS-101 10 mg/kg vs. SNS-101 30 mg/kg + anti-PD-1 1 mg/kg, $P = 0.0058$; SNS-101 30 mg/kg vs. SNS-101 30 mg/kg + anti-PD-1 1 mg/kg, $P = 0.0007$. **b** Terminal bleeds were collected from mice spun for serum, and cytokines/chemokines were measured by 31-plex, 23-plex and Th17 10-plex Luminex assays. Log-transformed data was analyzed by ANOVA with follow-up Tukey HSD testing. Heatmap was plotted using the "Pheatmap" R package and shows scaled cytokine/chemokine levels (Z-scores, scale, bottom right) within each cytokine across different treatment groups. Hierarchical clustering was performed by calculating the pairwise Euclidean distance between scaled cytokine levels of individual (group) samples and then merging into larger groups with the complete linkage algorithm until all samples merged into one final group. **c** Correlation between cytokines and tumor volume were generated using Pearson correlation coefficients. The top eight positively and negatively correlated cytokines are shown in the correlation matrix. **d** The fit of a linear regression model between the tumor volume and all cytokine/chemokine levels showed a strong relationship ($R^2 = 0.82$). Further recursive feature selection was performed with the Random Forest algorithm using "randomForest" and "caret" R packages. Selected cytokines were used to build a linear model to predict the tumor volume ($R^2 = 0.77$). The bar graph (left y-axis, top x-axis) shows the "importance" for model performance of the selected features. The line plot overlay (right y-axis, bottom x-axis) shows Root mean square error (RMSE) as a function of number of variables incorporated. The blue dot indicates the minimized RMSE of the key features (CCL7, IL-3, CXCL12, CCL24, CCL12, CCL2 and CXCL10). Source data are provided as a Source Data file.

protein was purified with a ProteinA affinity column (HiTrap™ MAb-Select SuRe™: Cytiva 1100349521) and stored in PBS, pH 7.4 buffer. The purity and identity of purified proteins were characterized using SDS-PAGE and Western blot using appropriate antibodies. Protein concentration of purified material was determined from the absorbance at 280 nm, where the molar extinction coefficient at 280 nm was calculated based on the amino acid sequence. Endotoxin levels were measured on an Endosafe nexgen-PTS instrument (Charles River). In vivo studies adhered to FDA guidance, maintaining endotoxin levels below 5EU/kg/h. For ex vivo CRS studies, endotoxin concentrations were kept below 0.01 EU/ml in blood, following Immuneed's specifications.

## VISTA:SNS-101 binding assays

SPR experiments were performed using a Bruker Sierra SPR-24 Pro instrument with Sierra Analyser 3.4.3. All SPR experiments were conducted at 25 °C using PBS containing 0.05% Tween 20 (Sigma Aldrich, Inc P1379) as running buffer at various pHs according to the goal of the experiment. To measure the binding affinity between VISTA and SNS-101, an IgG capture sensor chip (Bruker 1862623) was used to capture SNS-101 (ligand) on the sensor chip before flowing VISTA-His (analyte) over the surface. The stock solution (3–15 nM) of SNS-101, in experimental running buffer, was injected over an IgG sensor chip at 10 μL/min flow rate for 60–300 s in order to capture the antibody. A range of 300–500 RU was targeted for captured SNS-101 to achieve a final RU value between 60 and 120 range upon binding of the analytes. To determine the interaction with captured SNS-101, an 8–10 concentration series between 0 and 1960 nM VISTA-His were used in various assays. The highest concentration of analyte considered for measuring a particular binding interaction was determined from the affinity of the same interaction. VISTA-His or its variants were injected for 120–180 s at 45 μL/min flow rate and dissociation was monitored for 450 s to obtain the kinetic parameters. The IgG capture sensor chip was regenerated with 10 mM glycine (Bruker 1862654), pH 2.0 (25 μL/min for 60 s) after each complete injection to strip off any trace amounts of ligand and analyte from the chip before the next injection. All data were corrected by a double reference subtraction, where reference spot signal and blank injection (0 nM VISTA-His) were subtracted from each analyte response. For all antibodies, a 1:1 Langmuir binding, MTL (Mass Transport Limited) curve fitting model was used for data analysis and to obtain kinetic ($k_a$, $k_d$), and thermodynamic ($K_D$) parameters. To determine the pH dependence on kinetic and thermodynamic parameter of VISTA:SNS-101 interaction, the binding affinities were measured at various pH values between 5.8 and 7.4 with an increment of 0.2 pH unit (pH 5.8, 6.0, 6.2, 6.4, 6.6, 6.8, 7.0, 7.2, 7.4). The pH of the running buffer was adjusted by titration with 6 N HCl. Three different concentration series of VISTA-His were used for the entire pH range; 0–30 nM (pH 5.8, 6.0, 6.2), 0–240 nM (pH 6.4, 6.6, 6.8), 0-480 nM (pH 7.0, 7.2, 7.4) of VISTA-His. To determine the binding affinity between VISTA and PSGL-1, a shorter version of PSGL-1 with 19 N-terminal residues of human PSGL-1 fused to human IgG1-Fc[7] was prepared in-house. PSGL-1-19mer-Fc (7.5 nM) was used as a ligand and captured over IgG capture chip for 300 s with 10 μL/min flow rate. A concentration series of VISTA-His between 0 and 960 nM were injected over captured PSGL-1-19mer-Fc to measure the kinetic and thermodynamic dissociation constant between the two partners at pH 6.0 and 7.4.

## VISTA:SNS-101 Fab complex structure determination

To reduce glycosylation, a triple asparagine variant (N91Q/N108Q/N190Q) of VISTA was used for crystallization. This variant was expressed without a purification tag using the Expi293 expression system. The expressed protein was purified using IMAC (Ni-NTA) technique utilizing its large number of surface-exposed histidine residues and polished using anion exchange to remove minor impurities. Before crystallization, VISTA was deglycosylated using Endo Hf (New England Biolab P0703L). A complex of this purified and deglycosylated untagged VISTA and SNS-101 Fab was formed at pH 6.0 and purified using gel filtration chromatography. This procedure yielded a homogenous complex with a purity greater than 95% as judged from Coomassie stained SDS-PAGE. The concentration of the purified complex was adjusted to 23 mg/mL. Structure determination was performed at Proteros Biostructures GmbH. Crystals of the VISTA:Fab SNS-101 complex were grown in the following conditions: 0.1 M BIS-TRIS propane pH 8.25-8.75, 0.2 M NaBr, 16–24% PEG3350 at 293 K. Crystals appeared within 1 to 3 days in hanging drop vapor diffusion setups, were cryo protected with glycerol, and vitrified in liquid nitrogen. The X-ray diffraction data were collected at the SWISS LIGHT SOURCE (SLS, Villigen, Switzerland) using cryogenic conditions and the final data was processed to 2.59 Å resolution. The crystals belong to space group C 2. The data were processed using the programs autoPROC[54], XDS[55], and AIMLESS[56]. The VISTA:SNS-101 Fab complex structure was solved by molecular replacement using PHASER[57] with the published structure of VISTA (PDB-ID 6OIL) and the Fab model from the structure of a Fab:VISTA complex (PDB-ID 6MVL). Subsequent model building and refinement was performed according to standard protocols with COOT[58] and the software package CCP4[56], respectively (Supplementary Table 2). For the measure of the free R-factor, a measure to cross-validate correctness of the final model, about 4.9% of measured reflections were excluded from the refinement procedure. Several rounds of manual model building in COOT and bulk solvent correction, positional, and B-factor refinement using REFMAC[59] yielded the final model. The atomic coordinates and structure factors have been deposited in the RCSB Protein Data Bank, PDB entry 8TBQ.

## Binding and competition assays between VISTA and putative interaction partners

**ELISA Assays.** For recombinant proteins an ELISA-based binding assay was implemented measuring the interactions of hVISTA-Fc fusion protein with plate-bound human receptor proteins at pH 6.0 and pH 7.4. hVSIG-3-Fc (Acro Biosystems VS3-H5258), hVSIG-8-Fc (9200-VS), hPSGL-1-Fc (3345-PS), hSyndecan-2 (2965-SD) and hLRIG-1 (8504-LR; all R&D Systems) were coated in ELISA plates (Corning) at 5 µg/ml. Plates were blocked in 1% NFDM in PBS at pH 6.0 or 2% BSA in PBS at pH 7.4. For binding assays, C-terminal biotinylated hVISTA-Fc (R&D Systems AVI7126-050) was serially diluted in 1% NFDM in PBS at pH 6.0 or 2% BSA in PBS at pH 7.4 before being added to blocked plates. For the competition assay, SNS-101 or isotype control (Ultra-LEAF Purified Human IgG1 Isotype Control, BioLegend 403501) were serially diluted in 1% NFDM in PBS at pH 6.0 and pre-incubated with 1 µg/ml biotinylated hVISTA-Fc at pH 6.0 for 2 h at room temperature. After washing in PBS containing 0.05% Tween 20 at pH 6.0 or pH 7.4, HRP-labeled streptavidin (Thermo Fisher Scientific 21134) diluted 1:200 in PBS-NFDM at pH 6.0 or PBS-BSA at pH 7.4 was used as detection reagent.

**Klickmer™ assays.** To study the interaction between VISTA and native PSGL-1 on human T-cells, we employed flow cytometry assays. These assays utilized a phycoerythrin (PE)-labeled dextran polymer backbone, which carried a defined number of streptavidin domains loaded with biotinylated VISTA. To formulate the VISTA:Dextramer complex, Klickmer™-PE (Immudex DX01-PE) at 32 nM was combined with 100 nM biotinylated human VISTA (Acro Biosystems B75-H82E1) in pH 6.0 buffer (PBS + 0.5% BSA + 2 mM EDTA adjusted to pH 6.0 with 2-(N-morpholino)ethanesulfonic acid (MES) buffered saline (0.1 M MES, 0.9% NaCl$_2$, pH 4.7 (Thermo Fisher Scientific 28390)). This mixture was incubated for 30 min at room temperature in the dark. Prior to the assay, CD4$^+$ or CD8$^+$ T-cells (Lonza 4W-202 and 4W-302, respectively) were stimulated with Dynabeads human T-Activator CD3/CD28 (Thermo Fisher Scientific 1116D) for 3 days. For the assay, pre-assembled VISTA:Dextramer complex was incubated with SNS-101 or human IgG1 isotype control (BioLegend 403502) which were serially diluted in pH 6.0 buffer. As a control, Klickmer™-PE without VISTA was used. Post antibody-complex incubation, samples were added to 1.0 × 10$^5$ T-cells and incubated for 30 min at room temperature prior to washing in pH 6.0 buffer. Sytox Blue (Thermo Fisher Scientific) was added for live/dead cell discrimination and samples analyzed by flow cytometry to determine frequency of VISTA:Dextramer-PE positives. Representative flow cytometry data is shown in Supplementary Fig. 6. EC$_{50}$ values were calculated using GraphPad Prism software.

## Pharmacokinetics analysis

The study involving cynomolgus monkeys and the subsequent analysis of SNS-101 in their serum were conducted by Charles River. SNS-101 and h26A were administered to groups of cynomolgus monkeys (n = 4; 2/sex) once via intravenous infusion, using a temporary catheter, for a period of 1 h. Serum samples containing h26A were analyzed by ELISA on plates coated with hVISTA. Bound anti-VISTA mAb was detected using HRP-conjugated anti-human IgG1-Fc mAb (Bio-Rad HCA285P; 125-fold dilution in Blocking Buffer: 1% Fish Gelatin Blocking Agent (VWR 89411-096) in PBS) and HRP substrate TMB (SeraCare 5120-0050). The assay had a lower limit of h26A quantification (LLOQ) of 0.46 ng/ml. SNS-101 in cynomolgus serum was assessed by an electrochemiluminescent (ECL) assay on an MSD instrument (Meso Scale Diagnostics) using a custom-generated anti-ID antibody pair generated against SNS-101 (Bio-Rad AbD Serotec GmbH). Standard Meso Scale Discovery (MSD) 96-well plates were coated with 1 µg/mL AbD51857ad (capture antibody). The plates were blocked with 3% Bovine Serum Albumin (BSA) / Phosphate-Buffered Saline (PBS). SNS-101 bound to the plate was quantified by incubation with the detection antibody, AbD51841rao conjugated to Sulfo-TAG. ECL signal was detected using

MSD Sector Imager S 600 or SQ120 reader. LLOQ was 0.5 µg/mL. For mouse studies, SNS-101 concentrations in mouse serum were assayed using mouse anti-human IgG Fc (Abcam ab99757) immobilized in high bind microplates (Corning 2592) followed by detection using a peroxidase-conjugated mouse anti-human IgG F(ab)$_2$ fragment-specific reagent (Jackson Laboratories 209-035-097; 40,000-fold diluted in Blocking Buffer PBS + 2% BSA) and development using the HRP substrate TMB (Life Technologies 34028). The LLOQ for this assay was determined to 30 ng/mL. The concentrations in the samples were determined using non-linear regression with interpolation of unknown values from the prepared standard curve of mAbs using GraphPad Prizm 9 (GraphPad Software). Calculation of PK parameters using non-compartmental analysis was performed with Phoenix WinNonlin (version 8.3, Certara Corp.).

## CRS assays

HUVEC:PBMC co-culture assays were conducted essentially as described in ref. 22 using soluble antibody. HUVECs, allogenic to peripheral blood mononuclear cells (PBMCs) (Lonza C2519A), were expanded in Full EBM-2 Medium containing all BulletKit supplements (Lonza CC-3162). Cells were seeded into clear flat-bottom TC-treated 96-well plates (Fisher Scientific 3585) at a density of 30,000 cells/well in 100 µl medium. After culturing for 24 h, the medium was replaced with human PBMC's (Miltenyi 150-000-571, IQB IQB-PBMC102 and ATCC PCS-800-011) in full RPMI 1640 medium (Gibco A10491-01) containing 2% AB serum (Sigma H6914-100ML) and 1x non-essential amino acids (NEAA; Gibco 11140-050) (200 µl/well of 500,000 cells/ml). Antibodies diluted and titrated in full RPMI 1640 were added (100 µl/well) and the co-culture incubated at 48 h prior to cytokine analysis as explained below. The ex vivo circulating human blood assay was conducted by Immuneed AB. Briefly, fresh whole blood was taken from six healthy volunteers and a low amount of soluble heparin that allowed for analysis of drug-related effects on complement or coagulation cascade systems was added. The blood was immediately transferred to the ID.Flow test system, followed by administration of the test items, and set to circulate at 37 °C to prevent clotting. Samples were collected at baseline and at 4 h, then processed to plasma by centrifugation. They were subsequently stored at ≤−60 °C until cytokine analysis was conducted on the Multi-Array platform from Meso Scale Discovery (MSD).

## Mice

All mouse experiments were performed in specific-pathogen free facilities at genOway, Murigenics, or Washington University in St. Louis. Animals were housed in ventilated cage racks with micro-filtered tops and sterile bedding, n = 5/cage, at 21–24 °C and 50 ± 20% relative humidity, in rooms with at least ten room air changes per hour. Diurnal photoperiod; 12:12 light dark cycle. Water and food (rodent Maintenance Chow: Harlan Teklad product: 2018) given ad libitum.

## BRGSF-HIS mice

Experiments were performed at genOway S.A. Newborn (≤5 days of age) BALB/c Rag2$^{tm1Fwa}$Il2rγ$^{tm1Cgn}$Sirpα$^{NOD}$Flt3$^{tm1Irl}$ mice underwent intra-hepatic transplantation with ~0.9 × 10$^5$ human hematopoietic progenitor cells (hHPC; CD34+ cord blood cells) ~24 h after full body irradiation conditioning (2.8 Gy; X-ray source). Humanization rate and main immune cell proportions were evaluated in blood 12 weeks post-injection. BRGSF-HIS mice received four (4) intra-peritoneal injections every 2 to 3 days (D0, D + 2, D + 4 and D + 7) of 10 µg (in 150 µL of PBS 1X) of recombinant human Flt3L. At day 8, mice were injected via the tail vein with anti-CD3 (OKT3) or SNS-101 (VISTA) and blood was sampled at 2 h, 6 h, 24 h and 48 h post-injection. For each time point, at least 150 µL of whole blood was sampled per mouse through the jugular vein under gaseous anesthesia (isoflurane). Whole blood was kept at room temperature for 1 h to allow for coagulation and then centrifuged at 3000 g for 10 min at 4 °C. Supernatants containing sera

were collected into new tubes and stored at −80 °C until processing for ELISA. Mice were sacrificed at 24 h or 48 h post-injection and spleen were collected in FACS Buffer (PBS 1X, 3% FBS, 2 mM EDTA). Spleen was digested using spleen dissociation kit and GentleMACS Octo Dissociator with Heaters (Miltenyi Biotec) per manufacturer's instructions. Undigested tissues and debris were removed by filtering the cellular solution through a 70 μM filter in FACS Buffer. Cell number was evaluated using a Luna-FL™ automated cell counter (Logos Biosystems). For FACS analysis, 2 million cells were labeled with antibody cocktails (Supplementary Table 4) and incubated for 30 min at 4 °C in the dark. Cells were then washed in FACS Buffer before flow cytometry acquisition (Attune NxT, Thermo Fisher Scientific). Data analysis was performed using FlowJo (BD Biosciences) and GraphPad Prism software.

## In vivo efficacy in MC38 and MB49 syngeneic mouse tumor models

Single-sex cohorts (4–6 weeks old) were used to match origin sex of mouse cancer cell lines (MC38 (Kerafast ENH204-FP); female, MB49 (Millipore SCC148); male). For MC38, female human VISTA knock-in mice (C57BL/6N-Vsirtm1(huVSIR-ICP3, genOway; housed at Murigenics) were implanted with $1 \times 10^6$ MC38 cells/animal, and mice randomized into treatment groups as tumor volumes reached ~60–100 mm³. Mice were dosed intraperitoneally (i.p.) bi-weekly for 3 weeks with rat α-mPD-1 (RMP1-14; Bio X Cell BP0146) or rat IgG2a (2A3) isotype control (Bio X Cell BP0089) and thrice weekly for 3 weeks with α-hVISTA (SNS-101-m2) or mouse IgG2a (1.18.4) isotype control (Bio X Cell BP0085) at the dose levels indicated. Mice were euthanized with $CO_2$ and CDL. Terminal blood was collected in microtainer $K_2$EDTA tubes, kept on ice until centrifugation at 4 °C (within 10 min of collection), and plasma samples immediately stored at −80 °C. MC38 tumors were excised and placed in Hibernate-E (Thermo Fisher Scientific A1247601) or in 10% Neutral Buffered Formalin for 24 hours followed by transfer to 70% EtOH. For MB49, male hVISTA-KI mice were implanted with $1 \times 10^6$ cells. When tumor volumes reached ~60–100 mm³, mice were randomized into treatment groups. Antibodies were administered i.p.: rat α-mPD-1 (RMP1-14; Bio X Cell BP0146) or rat IgG2a (2A3) isotype control (Bio X Cell BP0089) bi-weekly, and α-hVISTA (SNS-101-m2) or mouse IgG2a (1.18.4) isotype control (Bio X Cell BP0085) thrice weekly, for 2 weeks with at the indicated doses. Tumor growth inhibition was calculated as previously described[60]. Significance was evaluated using Mann-Whitney unpaired $t$ test with $P < 0.05$ considered to be statistically significant.

## In vivo efficacy in MCA/1956 sarcoma mouse tumor model

Female hVISTA KI mice were subcutaneously inoculated with $1.5 \times 10^6$ MCA/1956 cells (Robert D. Schreiber Lab, Washington University, St. Louis; $n = 8$ mice/group). After 10 days, antibodies were administered i.p. bi-weekly: rat IgG2a (clone: 1-1, Leinco I-1177, 10 mg/kg) and human IgG1 isotype control Ab (Bio X Cell, BP0297, 20 mg/kg), anti-PD-1 (clone: RMP1-14, Leinco P372, 10 mg/kg) and human IgG1 isotype control Ab, rat IgG2a and SNS-101 Ab (20 mg/kg), or anti-PD-1 and SNS-101 (20 mg/kg). Tumor diameters were measured bi-weekly. Animals were euthanized when one diameter of a tumor reached 20 mm.

## In vivo efficacy in EG.7 mouse tumor model

EG.7-OVA cells ($1 \times 10^6$ cells per animal; ATCC CRL-2113) were injected subcutaneously into female hVISTA-KI mice (n = 8 mice/group). When tumors reached ~60–100 mm³, animals were randomized and treated thrice weekly for 3 weeks via i.p. injections of either isotype control IgGs, SNS-101-m2, or α-mouse CTLA-4 (clone 9H10, Bio X Cell BE0131). Tumor volumes were measured three times per week. Statistical significance in growth inhibition was determined using Mann-Whitney unpaired t-test, with $P < 0.05$ considered to be statistically significant.

## Analysis of tumor T-cell content

Single-cell suspensions were generated from tumor extracts through physical (gentleMACS Octo Dissociator, Miltenyi) and enzymatic (Liberase DL, Millipore Sigma LIBDL-RO) dissociation. Cell concentrations were determined and $1 \times 10^6$ cells per sample measured were subjected to viability (LIVE/DEAD™ Fixable Aqua Dead Cell Stain Kit, L34957), CD45-FITC (11-0451-82), CD4-Brilliant Violet™ 421 (404-0042-82) and CD8-PE (12-0081-82; all reagents from Thermo Fisher Scientific) staining in the presence of FcR blocking reagent (Biolegend 101320) in V-bottom, 96-well plates. The frequency of CD4$^+$ and CD8$^+$ T-cells was determined in the singlet, live, CD45$^+$ population by analytical flow cytometry analysis on a MACS Quant Analyzer 10 instrument (Miltenyi).

## Cytokine and chemokine profiling

Blood samples from mice were collected into microtainer $K_2$EDTA tubes at the end of study. Blood was kept on ice ≤10 min until centrifugation at 4 °C for 10 min at $1200 \times g$. The plasma samples were stored at −80 °C until analysis. Analytes were quantified using the following Bio-Rad assays according to manufacturer's instructions: Bio-Plex Pro Mouse Chemokine Panel, 31-plex, 12009159, Bio-Plex Pro Mouse Cytokine 23-plex Assay M60009RDPD and Bio-Plex Pro Mouse Th17 Cytokine 10-plex Assay 12010828. Detection was performed using a Bio-Rad Bio-Plex 200 instrument. For human cytokines from HUVEC:PBMC co-culture samples, we used the Bio-Plex Pro Human Cytokine 8-Plex Kit (Bio-Rad M50000007A). Mouse data were read into R and log-transformed to stabilize variances. For statistical analysis, ANOVA was used to compare means of different groups. Follow-up tests with Turkey HSD were carried out only when the ANOVA analysis showed a significance at α = 0.05. Cytokine heatmap showing the scaled cytokine/chemokine levels (Z-scores) within each cytokine across different treatment groups was plotted using the "Pheatmap" R package. Hierarchical clustering was performed by calculating the pairwise Euclidean distance between scaled cytokine levels of individual (group) samples and then merging closest ones into larger groups with the complete linkage algorithm. The clustering continued until all samples merged into one final group. HC was done in R with the "pheatmap" package. *Correlation analysis*: The correlation matrix among cytokines and tumor volume were constructed using the Pearson correlation. The visualization was done with the "corrplot" R package. Cytokines were ordered according to their correlation coefficients to the tumor volume, and top ten positively and negatively correlated cytokines were shown. *Recursive Feature Elimination (RFE)*: We first fitted a linear regression model between the tumor volume and all cytokine/chemokine levels, and it showed a strong relationship ($R^2 = 82\%$). We further ran recursive feature selection with the Random Forest algorithm using "randomForest" and "caret" R packages. Lastly, selected cytokines were used to build the linear model to predict the tumor volume.

## Analysis of binding profiles of SNS-101, VSTB174, h26A and hIgG isotype control to monocytes, neutrophils, NK cells and T cells

Anti-VISTA antibodies and isotype control (Human IgG1 Isotype control, Bio X Cell BP0297) were labeled with Alexa Fluor 647 using the protocol in the conjugation kit (Biotium). Monocytes, T-cells, and Neutrophils were labeled in a human blood sample (BIOIVT) by flow cytometry using CD45-VioGreen (130-110-638), CD3-PE-Vio 770 (130-113-140), CD16-Vio Bright B515 (130-119-616) and CD14-PE (130-113-147) in the presence of FcR Blocking Reagent (130-059-901; all reagents from Miltenyi Biotec and antibodies diluted 50-fold as per manufacturers recommendation). Samples with appropriate FMO (Fluorescence Minus One) controls for each antibody were analyzed in parallel. RBC were lysed using Lysis Buffer (BD Biosciences 555899). Samples were washed in 1x PBS pH7.4 containing 1% heat inactivated fetal bovine serum, and propidium iodide (PI) staining (Miltenyi 130-093-

233) for Live/dead cell discrimination done immediately before analysis using a MACS Quant Analyzer (Miltenyi). For NK cells, isolated human NK cells (Hemacare/Charles River Labs) were stained with labeled anti-VISTA or isotype control mAbs and CD56-FITC (Miltenyi 130-114-740; 50-fold diluted) in MACSQuant Running Buffer (Miltenyi 130-092-747), and Sytox Blue (Thermo Fisher Scientific S34862) was used for gating for live cells.

## Statistics and reproducibility

No statistical methods were used to predetermine sample sizes. Investigators were not blinded to allocation during experiments and outcome assessment; however, in vivo experiments were conducted at independent contract research organizations with no interests in experimental outcomes. Sample sizes for experiments involving animals subjects were chosen according to 3R guidance for responsible use and animal welfare. Statistical analyses were conducted using GraphPad Prism (versions 8–10) and R, with specific methods detailed in the figure legends and above where applicable. No data were excluded from the analyses.

## Reporting summary

Further information on research design is available in the Nature Portfolio Reporting Summary linked to this article.

## Data availability

Protein structure data has been deposited at RCSB with the PDB identifier 8TBQ. Images (raw data) were deposited at https://doi.org/10.5281/zenodo.10048138. Source data are provided with this paper. All other data that support the findings of this study are available from the corresponding author upon reasonable request. Source data are provided with this paper.

## Code availability

No unique code was developed in this work. R scripts used for analyses are available at https://github.com/SenseiBio/SenseiBio or from the corresponding author upon reasonable request.

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

## Author contributions

Conceptualization: T.T., R.D.S., E.H.vdH; Methodology: T.T., A.M., Y.K., F.F., Z-G.J., T.E., K.M., Z.B., A.O., F.F., A.C., G.Z., G.H.M., Y.T., K.T., R.D.S., E.H.vdH; Investigation: A.M., Y.K., Z-G.J., T.E., K.M., Z.B., A.O., F.F., A.C., G.Z., G.H.M., Y.T.; Analysis: T.T., F.D.S., A.M., Y.K., F.F., Z-G.J., T.E., K.M., Z.B., A.O., F.F., A.C., G.Z., G.H.M., Y.T., K.T., R.D.S., E.H.vdH; Visualization: T.T., F.D.S., A.M., Y.K., F.F., G.H.M., Y.T., E.H.vdH; Writing—original draft: T.T., F.D.S., A.M., G.Z., E.H.vdH; Writing—review and editing: G.H.M., Y.T., K.T., R.D.S.; Writing—revision: T.T., F.D.S., E.H.vdH; Supervision: T.T., K.T., R.D.S., E.H.vdH.

## Competing interests

T.T., F.D.S., A.M., Y.K., F.F., Z-G.J., T.E., K.M., Z.B., A.O., F.F., A.C., G.Z., and E.H.vdH are current or former employees of Sensei Biotherapeutics, Inc. G.H.M., and K.T. are employees of genOway. Y.T. has no competing interests to declare. R.D.S. is a member of the Scientific Advisory Board of Sensei Biotherapeutics, Inc.
