## [Peer Review File · Nature Communications]

VISTA checkpoint inhibition by pH-selective antibody SNS-101 with optimized safety and pharmacokinetic profiles enhances PD-1 responseREVIEWER COMMENTS

Reviewer #1 (Remarks to the Author):

VISTA plays an important role in maintaining myeloid and T cell quiescence and has emerged as a potential target for cancer immunotherapy, especially in combination with other immune checkpoint inhibitors. Previous VISTA antibodies have issues of rapid clearance and cytokine release syndrome (CRS). Authors developed a PH-selective VISTA antibody, SNS-101, that exhibits a better PK and safety profile but less CRS risk. In addition, authors showed that SNS-101 sensitizes MC38 (colon cancer) and 1956 cells (sarcomas) to anti-PD-1 therapy in mice. Currently, SNS-101 is in phase I clinical trials for patients with advanced solid tumors. Overall, this manuscript presents compelling evidence supporting the potential of SNS-101 as a promising candidate for cancer immunotherapy, warranting publication in Nature Communications if the following points are addressed.

1. Expanded efficacy assessment: Given that SNS-101 will be used to treat patients with advanced solid tumors, it is important to determine SNS-101 efficacy in various tumor types, such as pancreatic, liver, lung, ovarian cancer etc. Even negative data will be valuable and informative in this context.
2. Figure 6a-f, it is better to include an end-point bar or dot plot to compare tumor volumes with statistics.
3. Figure 7 seems irrelevant. It is not clear how this proteomics analysis contributes to the central theme of the manuscript.

Reviewer #2 (Remarks to the Author):

The results are very important for the development of new antibodies and a safer therapy. The antibody presented looks promising, but the authors should consider using other reference antibodies for their next study. While the SNS-101 antibody blocks the VISTA-PSGL-1 pathway, the JNJ antibody, which blocks the VISTA-VS1G-3 pathway, is used as a reference. The authors should instead explore antibodies that inhibit the VISTA - PSGL-1 interaction. For example, they could consider using the antibodies presented by Johnson et al. in 2019 (Nature 2019 Oct;574(7779):565-570. doi: 10.1038/s41586-019-1674-5) or Mehta et al. in 2020 (Sci Rep. 2020; 10: 15171. doi: 10.1038/s41598-020-71519-4).

The following should be added to the results section:

- a) Competition ELISA of JNJ and h26A antibodies (as shown in Fig. 2b-f for SNS-101)
- b) For Fig 6a-f, an additional figure depicting the average cancer volume in the groups.

The following should be added to the methods section:

- a) A sequence of the SNS-101 antibody.
- b) The catalogue numbers of the chemicals used.

Additional questions:

- 1) Are the authors sure that the untagged VISTA was purified by NI-NTA?
- 2) What could cause such a rapid decrease in h26A concentration in the serum of NHPs?
- 3) Why were only three doses used for the in vivo cytotoxicity test?
- 4) Why were the PK (pharmacokinetics) for h26A (ELISA) and SNS-101 (electrochemiluminescence assay) not performed with the same method?

Reviewer #3 (Remarks to the Author):

The authors generate and extensively characterize a novel pH-selective VISTA antibody. Since the first pH-selective VISTA antibody was described by Johnston et al, a number of additional VISTA/receptor interactions have been described and the authors examine all five interactions as well as five other published VISTA antibodies. They show very favorable safety and pharmacokinetic properties such as long half-life, failure to bind to the large pool of VISTA positive cells in vivo at pH7.4 but binding in an acidic tumor microenvironment. In addition, they show their VISTA mab does not induce cytokine release syndrome in several in vitro and in vivo models, while a non-pH-selective mab does. They show safety in non-human primate studies. They show in vivo anti-tumor efficacy in 2 mouse tumor models in combination with anti-PD-1. They show the in vivo efficacy of SNS-101 and anti-PD-1 correlates with changes in key cytokine and chemokine networks. They use VISTA-Fc to transduce a signal into PSGL-1 positive human CD4 T cells and do a proteomic analysis. They identify a number of changes in pathways and emphasize the IL-7 and TSLP pathways but do not confirm their involvement. The binding and tumor therapy studies are similar to previous work but the CRS, pharmacokinetic, and mechanism studies are novel. The studies provide strong justification for clinical testing of their pH-selective VISTA antibody.

1. Need affiliations of most authors.
2. In abstract "SNS-101 demonstrated in vivo efficacy," say in what model/species.
3. Ref 12 Curis. (Curis, Inc., <https://investors.curis.com/events-and-presentations?item=100,2022>). Was not active. Check that ref engages an active site.
4. "SNS-101, a fully human monoclonal IgG1 antibody (mAb) specific for the protonated (active) form of VISTA," at this point, say whether it also recognizes mouse and NHP VISTA.
5. "previously observed issue of TMDD associated with non-pH-selective VISTA antibodies has been completely resolved". Soften if not in human clinical trial.
6. Invert 1d so it is same orientation as 1c.
7. Would be useful to add a panel in figure 2 showing ratio of binding at pH6 vs 7.4 for each interaction.
8. "Only PSGL-1 and Syndecan-2 displayed a significant dose-dependent and saturable interaction at pH 6.0, but not at pH 7.4." Fig 2a certainly shows PSGL-1 and Syndecan have the most pH dependence but all the ligands show a large pH shift. None are clearly saturable. Clarify.
9. In 2 g,h, the binding is said to be to PSGL-1 but T cells may express other of the ligands? CRISPR out PSGL-1 in human T cells or soften the statement.
10. In fig 3 say whether mab is soluble or coated since their ref 24 uses coated.
11. "JNJ dose-dependently induced cytokine release and short-term monocyte activation, accompanied by decreased cell proportions (G. M., Z. B., A. C., A. M., T. T., E. H., K. T.; submitted). Show data or remove
12. Specify number of repeats for each experiment.
13. I am skeptical of the data of Figure 4K, L. The data in 4K is significant only because the %CD86 declines in the PBS treated mice between 24 and 48 hr, a hard to understand result. In 4L, the summation of CD4 and CD8 in CD3+ is sometimes below 15%, a hard to understand result. Unless these have been repeated and an explanation can be provided in a rebuttal letter, remove 4K, L. In 4L labeling, do you mean %CD4/CD8 in hCD3 ? (CD4 or CD8 in hCD3 ?)
14. "PK profile of SNS-101 in ... in wildtype (WT) C57BL6/J mice." Say whether SNS-101 reacts with mouse VISTA.

15. "we assessed the PK profile of SNS-101 in NHPs". Say whether SNS-101 and h26A react with NHP VISTA.
16. Say that PD-1 mab dose of 1 mg/kg is about 1/10 of often used dose.
17. In the discussion, the authors say "key changes correlating with inhibition of tumor growth include the downregulation of CCL-2, CCL-7, CCL-12, and the upregulation of CXCL-10, CCL-24, CXCL-12 and IL-3." I find this explanation to be clear; however, elsewhere in the results they express this as positive and negative correlations, which is less easily understood. I suggest using the above explanation throughout.
19. Suggest Ext Figure 8 (1956 sarcoma model, be included in main figures; also say this uses standard amount of PD-1 mab).
20. "Endotoxin level was measured on an Endosafe nexgen-PTS instrument (Charles River) as needed." Specify amount acceptable for studies.
21. "PSGL-1-19mer-Fc was co-expressed with plasmids encoding glucosaminyl (N-acetyl) transferase (core 2, GCNT1) and alpha (1,3)-fucosyltransferase-7 enzymes (Fut3) at 2:1:1 ratio to enhance the post-translational modification (sialyl-Lewis X)." Specify the 19 amino acids as there is ambiguity. what cell? Transient or stable? Johnston reported the sulfates were important, not sialyl-Lewis X. Was the PSGL-1-19mer-Fc further purified to enrich for sulfation as done in some of Johnston's studies.
22. "All SPR experiments were conducted at 25 °C using PBS containing 0.05% Tween 20 as running buffer at various pHs". Confirm in PBS even for pH 5.8. no MES?
23. "For VISTA interacting with native PSGL-1 on human T-cells, flow cytometry assays were conducted using dextramers loaded with biotinylated VISTA." Describe loading. Show example FACS in Ext data.
24. "diluted in MES buffer pH 6.0". specify concentrations, constituents.
25. In the HUVEC:PBMC co-culture assays, specify the PBMC are allogeneic to HUVEC.
26. "1024 101 Ab three times every 3 days". Ambiguous, clarify
27. Line 234. Specify tumor model.
In MC38 methods, specify mab doses.
28. Is a human IgG1/mouse IgG2a needed or does a non-depleting Fc work in tumor immunotherapy experiments? Does any data inform whether the mechanism is blocking, depletion, or transduction of a VISTA signal?
29. "were labelled with AF-674". Do you mean Alexa647?
30. "line 1045. CD4+ T-cells treated with or without VISTA-Fc". Provide details of method including Fc isotype, concentration, soluble or plate-bound, time, cells & culture conditions, any other signals (CD3, CD28?), etc. Justify not using an Fc control as FcReceptors transduce signals. If no isotype control was used, say this as a weakness of study.
31. In Supp Fig 1, show data as raw histograms of isotype control and stained cells.
32. Line 262. Discuss/reference why this is an M1-like set of cytokines.
33. Line 275. "Among all enriched pathways, 76 (14.4%) correspond to previous findings from functional and transcriptomic studies?" from VISTA studies? Reference

34. The emphasis on IL-7 and TSLP pathways is poorly justified. They are presented as if they are for sure the important pathways. Other pathways, higher on the list, such as downstream TCR signaling, would seem more likely to be important. Soften the emphasis on IL-7 and TSLP pathways unless a stronger justification can be provided, note that they need experimental validation.

35. Does the proteomics analysis include phospho-peptides?

Reviewer Comments

Reviewer 1

We would like to express our gratitude to Reviewer 1 for the insightful comments and suggestions. We believe that addressing these points has significantly improved the quality of our manuscript.

Comment/Question	Response	Line Number
1. Expanded efficacy assessment: Given that SNS-101 will be used to treat patients with advanced solid tumors, it is important to determine SNS-101 efficacy in various tumor types, such as pancreatic, liver, lung, ovarian cancer etc. Even negative data will be valuable and informative in this context.	We are grateful for the reviewer's astute recommendation. Recognizing the importance of assessing SNS-101's efficacy in a spectrum of tumor types, especially given its potential application in treating patients with advanced solid tumors, we have expanded our investigations. Specifically, we have incorporated studies on bladder (MB49) and lymphoma (EG.7) tumor models. Additionally, we have enriched the data from the 1956 sarcoma experiment, where the primary endpoint was response/survival, and this is now integrated into the main figure. The outcomes of these expanded studies are detailed in Fig. 6c, e, f and Supplementary Fig. 11. It's noteworthy that while SNS-101 showed encouraging results in the MB49 model, mirroring the findings in the MC38 model, it did not manifest significant anti-tumor activity in the EG.7 model, especially when juxtaposed with anti-CTLA-4. These findings underscore the nuanced therapeutic potential of SNS-101 in cancer immunotherapy and emphasize the importance of context in evaluating its efficacy.	291-318
2. Figure 6a-f, it is better to include an end-point bar or dot plot to compare tumor volumes with statistics.	We appreciate the feedback on Fig. 6a-f. In response, we have revised Fig. 6 and included end-point dot plots to provide a clearer comparison of tumor volumes (Fig. 6b for MC38 and Fig. 6d for MB49). The updated Fig. 6 includes the relevant statistical analysis.	
3. Figure 7 seems irrelevant. It is not clear how this proteomics analysis contributes to the central theme of the manuscript.	We thank the reviewer for pointing out the concerns regarding Fig. 7. The intention behind including the proteomics analysis was to reveal pathways responsive to VISTA:PSGL-1 engagement in primary human CD4⁺ T-cells. However, we understand the reviewer's perspective on its relevance to the central theme. Based on the	

	feedback, we have decided to remove the data and the figure.	
--	--	--

Reviewer 2

We are grateful to the reviewer for recognizing the significance of our findings and for the constructive feedback provided. We concur with the suggestion to explore other reference antibodies that specifically target the VISTA-PSGL-1 interaction. It is worth noting that we have already illustrated the epitopes on VISTA that are targeted by various anti-VISTA antibodies, including those mentioned, in our Fig. 1 (j and h, respectively). While the primary focus of this manuscript was to differentiate non-pH sensitive, clinical-stage antibodies, we are poised to integrate these additional antibodies in our forthcoming research endeavors. This will undoubtedly provide a more holistic comparison and bolster the rigor of our investigations. We sincerely value this insightful recommendation.

Comment/Question	Response	Line Number
The following should be added to the results section: a) Competition ELISA of JNJ and h26A antibodies (as shown in Fig. 2b-f for SNS-101) b) For Fig 6a-f, an additional figure depicting the average cancer volume in the groups.	Thank you for these suggestions. a) In response to the suggestion, we have incorporated competition ELISA data for both JNJ and h26A antibodies, analogous to the presentation for SNS-101 in Fig. 2b-f. This new data can be found in Fig. 2g-k. By doing so, we believe it facilitates a more direct and comprehensive comparison among the antibodies. b) We have undertaken a thorough revision of Fig 6. As recommended, the figure now showcases the average tumor volume for each group. Additionally, to enhance clarity and provide a more detailed perspective, we have incorporated an end-point dot plot, which allows for a more lucid comparison of tumor volumes (refer to Fig. 6b for MC38 and Fig. 6d for MB49). The revised Fig. 6 is also complemented with the pertinent statistical analyses, ensuring that readers gain a holistic and clear understanding of the treatment effects observed across the various in vivo models.	171-177 291-318
The following should be added to the methods section: a) A sequence of the SNS-101 antibody. b) The catalogue numbers of the chemicals used.	We appreciate the attention to detail: a) We have now included the patent application US20230272082A1 in the Methods section to provide clarity on the sequence of SNS-101. b) Additionally, we have added the catalogue numbers for all the chemicals used in our study to the Methods section.	931

Additional questions: 1) Are the authors sure that the untagged VISTA was purified by NI-NTA?	Yes, we are confident in our purification method. Specifically, untagged VISTA, which has 14 surface histidine residues, binds to Ni-NTA. We purified it from CHO cell culture supernatants using Ni-NTA and ion exchange chromatography. After purification, we deglycosylated the untagged VISTA with the Endo Hf enzyme (New England Biolab – P0703L). We then prepared a complex of deglycosylated VISTA and SNS-101 Fab at pH 6.0. To ensure we used only the bound form for crystallization, we further purified this complex using gel filtration chromatography. Our SDS-PAGE analysis confirmed that this procedure yielded a complex with a purity exceeding 95%.	
2) What could cause such a rapid decrease in h26A concentration in the serum of NHPs?	The rapid decline in h26A concentration observed in NHP serum can be attributed to target-mediated drug disposition (TMDD). TMDD is a nonlinear pharmacokinetics (PK) phenomenon resulting from the high-affinity binding of a drug to its pharmacological target, exemplifying the interplay between pharmacodynamics (PD) and PK (An G. J Clin Pharmacol. 2020; 60: 149-163). VISTA is present in intracellular endosomes and undergoes recycling between the cell surface and endosomes during endosomal trafficking (Patent WO2018169993A1). The binding affinity of an anti-VISTA antibody to VISTA at acidic pH is crucial for its retention during this trafficking process. When a non-pH-sensitive antibody engages VISTA, the resulting antibody-VISTA complex is internalized. If these antibodies exhibit reduced binding to VISTA at acidic pH, they may dissociate from VISTA during the recycling process, leading to their entrapment or degradation within cells. This results in suboptimal target engagement and continuous depletion of circulating antibodies, an observation also reported by Johnston et al.	
3) Why were only three doses used for the in vivo cytotoxicity test?	We apologize for any oversight. For the in vivo studies depicted in Fig. 6, SNS-101 was administered thrice weekly throughout the study duration. Specifically, in the MC38 and MB49 tumor models, SNS-101 was administered a total of eight times.	775

4) Why were the PK (pharmacokinetics) for h26A (ELISA) and SNS-101 (electrochemiluminescence assay) not performed with the same method?	The methods chosen for PK analysis were tailored to the unique properties of each antibody: a) h26A: Given its ability to bind VISTA at neutral pH, we employed an antigen-capture ELISA assay. This method captures h26A from cyno serum using VISTA-coated plates, capitalizing on its specific binding properties. b) SNS-101: Since SNS-101 does not bind VISTA at neutral pH—and consistently adjusting the pH of serum samples is not feasible—we developed an anti-idiotypic antibody pair highly specific for SNS-101. This was used in an electrochemiluminescent assay format on an industry-standard platform (MSD) that has been validated for clinical sample analysis.	
--	---	--

Reviewer 3

Comment/Question	Response	Line Number
1. Need affiliations of most authors.	Thank you for pointing this out. We have updated the manuscript to include the affiliations of all authors.	4-10
2. In abstract “SNS-101 demonstrated in vivo efficacy,” say in what model/species.	We appreciate the feedback. We have now specified in the abstract that the in vivo efficacy of SNS-101 was demonstrated in mouse tumor models.	27
3. Ref 12 Curis. (Curis, Inc., https://investors.curis.com/events-and-presentations?item=100 , 2022). Was not active. Check that ref engages an active site.	We apologize for the oversight. We have checked and updated the reference to ensure it directs to an active and relevant site.	
4. “SNS-101, a fully human monoclonal IgG1 antibody (mAb) specific for the protonated (active) form of VISTA,” at this point, say whether it also recognizes mouse and NHP VISTA.	Thank you for the suggestion. We have clarified in the manuscript that SNS-101 cross-reacts with NHP, but not mouse VISTA.	69; 259
5. “previously observed issue of TMDD associated with non-pH-selective VISTA antibodies has been completely resolved”. Soften if not in human clinical trial.	We understand the concern. We have revised the statement to: "... in a preclinical setting, we preclinically demonstrate the previously observed issue of TMDD associated with non-pH-selective VISTA antibodies has been completely resolved,... Current clinical trials aim to further validate this result in humans.”	64-67
6. Invert 1d so it is same orientation as 1c.	Thank you for the suggestion. We have adjusted the orientation of Fig. 1d to match that of Fig. 1c for consistency.	
7. Would be useful to add a panel in figure 2 showing ratio of binding at pH6 vs 7.4 for each interaction.	That's a valuable suggestion. We have added a new panel in Fig. 2a to depict the ratio of binding at pH 6.0 versus pH 7.4 for each interaction, providing a clearer visualization of the pH-dependent binding differences.	166-167
8. “Only PSGL-1 and Syndecan-2 displayed a significant dose-dependent and saturable interaction at pH 6.0, but not at pH 7.4.” Fig 2a certainly shows PSGL-1 and Syndecan have the most pH dependence but all the ligands show a large pH shift. None are clearly saturable. Clarify.	We appreciate the reviewer's observation. We acknowledge that the term “saturation” might have been used prematurely. Our intention was to highlight that, at high protein concentrations at pH 6, the binding curves for PSGL-1 and Syndecan-2 begin to level off, with OD ₄₅₀ values reaching 2.5 and above, suggesting a trend towards saturation. To provide clearer evidence, we have added a figure that contrasts the binding at pH 6 vs. 7.4 for the five receptors (Fig. 2a). This emphasizes the pronounced pH-selective binding of PSGL-1 and Syndecan-2 relative to the other	164-167

	receptors. We have revised the manuscript to better reflect this and address the reviewer's concerns.	
9. In 2 g,h, the binding is said to be to PSGL-1 but T cells may express other of the ligands? CRISPR out PSGL-1 in human T cells or soften the statement.	Expression analysis of PSGL-1, Syndecan-2, LRIG-1, VSIG-3 and VSIG-8 on primary T-cells (activated and cultured as in original Fig. 2g & h) by flow cytometry (Supplementary Fig. 13) shows only PSGL-1 and to a lesser extent LRIG-1 to be expressed. Syndecan-2, VSIG-3 and VSIG-8 were not detected. Absence of VSIG-3 on T-cells is consistent with previous report (Johnston et al.). Additionally, Johnston et al. did show that knockout of PSGL-1 in T-cells lead to loss of binding of recombinant VISTA at low pH. Given these findings, we believe that PSGL-1 is the primary ligand responsible for the observed binding in Fig. I, m.	178-182; 399-400
10. In fig 3 say whether mab is soluble or coated since their ref 24 uses coated.	Thank you for pointing this out. In Fig. 3, the mAbs used were soluble. We have updated the figure legend and the corresponding Methods section in the manuscript to specify this detail.	724; 1106
11. "JNJ dose-dependently induced cytokine release and short-term monocyte activation, accompanied by decreased cell proportions (G. M., Z. B., A. C., A. M., T. T., E. H., K. T.; submitted). Show data or remove	We understand the reviewer's request for clarity. Given the constraints of the current manuscript, we have opted to remove this statement. We will present the detailed data in a subsequent publication.	
12. Specify number of repeats for each experiment.	We apologize for the oversight. We have now specified the number of repeats for each experiment in the respective figure legends.	
13. I am skeptical of the data of Figure 4K, L. The data in 4K is significant only because the %CD86 declines in the PBS treated mice between 24 and 48 hr, a hard to understand result. In 4L, the summation of CD4 and CD8 in CD3+ is sometimes below 15%, a hard to understand result. Unless these have been repeated and an explanation can be provided in a rebuttal letter, remove 4K, L. In 4L labeling, do you mean %CD4/CD8 in hCD3 ? (CD4 or CD8 in hCD3 ?)	We appreciate the reviewer's careful examination of Fig. 4k and 4l. To address the concerns:  1. Regarding the summation of CD4⁺ and CD8⁺ T-cells being below 15%: This observation is specific to the OKT3 treated group. This result aligns with the known depleting effect of OKT3 on T-cells. As supported by literature, OKT3 has been shown to mediate T-cell cytolysis, which can explain the observed reduction (Wong JT, Eylath AA, Ghobrial I, Colvin RB. "The mechanism of anti-CD3 monoclonal antibodies. Mediation of cytolysis by inter-T cell bridging." Transplantation. 1990 Oct;50(4):683-9). 2. Concerning the axis labeling in Fig 4l: We acknowledge that the labeling could have been clearer. The data was indeed intended to represent "% CD4⁺ or CD8⁺ in hCD3⁺." To enhance clarity, we have now separated this graph into individual analyses for CD4⁺ and CD8⁺ populations. 	241-248

	3. Addressing the observations in Fig 4k: We concur that the difference between the PBS and OKT3 groups wasn't statistically significant, likely due to fewer data points and increased variability in the PBS samples. However, a discernible trend exists, with OKT3 showing an expected increase in CD68+ monocytes. We have revised the language pertaining to the results in Fig 4K in light of your feedback and relocated panel 4k to Supplementary Fig. 10 for further clarity.	
14. "PK profile of SNS-101 in ... in wildtype (WT) C57BL6/J mice." Say whether SNS-101 reacts with mouse VISTA.	Thank you for raising this point. SNS-101 does not cross-react with mouse VISTA. We have added this information to the relevant section in the manuscript for clarity.	69; 259
15. "we assessed the PK profile of SNS-101 in NHPs". Say whether SNS-101 and h26A react with NHP VISTA.	We apologize for the oversight. Both SNS-101 and h26A do cross-react with NHP VISTA. This information has now been included in the manuscript for clarity.	269
16. Say that PD-1 mab dose of 1 mg/kg is about 1/10 of often used dose.	We appreciate the suggestion. We have added a statement in the relevant section specifying that the PD-1 mAb dose of 1 mg/kg used in our study is approximately one-tenth of the commonly used dose.	299
17. In the discussion, the authors say "key changes correlating with inhibition of tumor growth include the downregulation of CCL-2, CCL-7, CCL-12, and the upregulation of CXCL-10, CCL-24, CXCL-12 and IL-3." I find this explanation to be clear; however, elsewhere in the results they express this as positive and negative correlations, which is less easily understood. I suggest using the above explanation throughout.	We appreciate the feedback on the clarity of our explanations. We understand that the terms 'positive' and 'negative' correlations can be ambiguous in the context of our results. To maintain consistency and clarity throughout the manuscript, we will adhere to more explicit terminology such as 'decrease'/'increase' and 'downregulation'/'upregulation'. This change will be made in all relevant sections of the results and discussion to ensure that our findings are easily understood.	346; 447-448
18. CXCL-12 and IL-3." I find this explanation to be clear; however, elsewhere in the results they express this as positive and negative correlations, which is less easily understood. I suggest using the above explanation throughout.	Please see above.	346; 447-448
19. Suggest Ext Figure 8 (1956 sarcoma model, be included in main figures; also say this uses	We appreciate the suggestion. Ext. Data Fig. 8, showcasing the 1956 sarcoma model, has now been incorporated into the main figures, along	311-315

standard amount of PD-1 mAb).	with corresponding spider plots of tumor growth, as Fig. 6e, f. Additionally, we have clarified in the figure legend and accompanying text that a standard amount of PD-1 mAb was used in this model.	
20. "Endotoxin level was measured on an Endosafe nexgen-PTS instrument (Charles River) as needed." Specify amount acceptable for studies.	We have ensured that the endotoxin levels adhered to the recommended guidelines. Specifically, for in vivo studies, the endotoxin levels were maintained below 5EU/kg/h, in accordance with the FDA guidance. For ex vivo CRS studies, the endotoxin levels were kept below 0.01 EU/ml as the final concentration in blood, as specified by Immuneed. For further reference, the FDA guidance document on pyrogen and endotoxins testing (https://www.fda.gov/regulatory-information/search-fda-guidance-documents/pyrogen-and-endotoxins-testing-questions-and-answers).	966-969
21. "PSGL-1-19mer-Fc was co-expressed with plasmids encoding glucosaminyl (N-acetyl) transferase (core 2, GCNT1) and alpha (1,3)-fucosyltransferase-7 enzymes (Fut3) at 2:1:1 ratio to enhance the post-translational modification (sialyl-Lewis X)." Specify the 19 amino acids as there is ambiguity. what cell? Transient or stable? Johnston reported the sulfates were important, not sialyl-Lewis X. Was the PSGL-1-19mer-Fc further purified to enrich for sulfation as done in some of Johnston's studies.	We appreciate the detailed feedback. The 19 amino acid sequence from hPSGL1 utilized in PSGL1-19mer-Fc is QATEYEYLDYDFLPETEPP. This sequence is pivotal for binding both P-selectin (https://doi.org/10.1016/0092-8674(95)90173-6), and hVISTA (https://doi.org/10.1038/s41586-019-1674-5). For expression, we employed the Expi293 expression system, specifically using the Expi293F transient cell line, which is a modified variant of HEK293 cells. Our aim in incorporating the sialyl-Lewis X modification to PSGL-119mer-Fc was to maintain the highest resemblance to the native system. In our assays (as shown in Supplementary Fig. 5), PSGL-1-19mer Fc was purified using protein A affinity resin. We did not perform any additional ion exchange purification to enrich the sulfated PSGL-1-19mer Fc.	
22. "All SPR experiments were conducted at 25 °C using PBS containing 0.05% Tween 20 as running buffer at various pHs". Confirm in PBS even for pH 5.8. no MES?	Indeed, all SPR experiments were conducted using only PBS buffer across the entire pH range, including pH 5.8. We opted for PBS to circumvent any buffer mismatch effects. PBS is recommended for this pH range as per standard protocols (https://cshprotocols.cshlp.org/content/2006/1/pdb.rec8543.full).	
23. "For VISTA interacting with native PSGL-1 on human T-cells, flow cytometry assays were conducted using dextramers loaded with biotinylated VISTA." Describe loading. Show example FACS in Ext data.	We have elaborated on the dextramer loading procedure in the Methods section for a more comprehensive understanding. Additionally, to provide a visual representation, we have included exemplary FACS data in Supplementary Fig. 7.	1054-1075; Supp. Fig. 7
24. "diluted in MES buffer pH 6.0". specify concentrations, constituents.	The Method section has been updated to specify the concentrations and constituents of the pH 6.0 buffer used in this study.	

25. In the HUVEC:PBMC co-culture assays, specify the PBMC are allogeneic to HUVEC.	In the HUVEC:PBMC co-culture assays described in the Method section, we have now clarified that the PBMCs were allogenic to the HUVECs.	1106-1107
26. "1024 101 Ab three times every 3 days". Ambiguous, clarify	In the Methods section, we have clarified the dosing regimen for SNS-101. The treatment was administered over the course of the study (3 weeks), with each dose given thrice per week.	
27. Line 234. Specify tumor model. In MC38 methods, specify mab doses.	For the tumor model mentioned on Line 234, we have now specified that it is the "MC38 colon carcinoma model". In the MC38 Methods section, the doses of antibodies administered were 1mg/kg for anti-mPD-1 and 10 mg/kg or 30 mg/kg of SNS-101-m2 post-randomization". This has been clarified in the revised manuscript.	1156-1165
28. Is a human IgG1/mouse IgG2a needed or does a non-depleting Fc work in tumor immunotherapy experiments? Does any data inform whether the mechanism is blocking, depletion, or transduction of a VISTA signal?	In prior research, it was established that the Fc-engagement of a clinical non-pH-selective anti-VISTA IgG1 antibody, which is effector-competent, with FcγR on myeloid cells prompted the secretion of cytokines and chemokines. This secretion is pivotal for the complete trans-activation of T-cells, culminating in notable anti-tumor efficacy in vivo . Such outcomes were evident with the antibody JNJ-61610588, as referenced in the patent WO2017175058A1 and presented in the study "A human anti-VISTA antibody induces antitumor responses via a unique mechanism of action" (https://webcast.aacr.org/console/player/31334?mediaType=slideVideo&). When substituting IgG1 with an IgG4 framework, which lacks effector functions, there was a marked decrease in cytokine secretion from myeloid cells and a subsequent decline in anti-tumor activity. Given these observations, an Fc-competent framework, such as Human IgG1 or mIgG2a, is imperative for achieving the intended therapeutic results.	
29. "were labelled with AF-674". Do you mean Alexa647?	Yes, the correct description is "Alexa Fluor 647". This has been clarified in the revised manuscript.	1249
30. "line 1045. CD4+ T-cells treated with or without VISTA-Fc". Provide details of method including Fc isotype, concentration, soluble or plate-bound, time, cells & culture conditions, any other signals (CD3, CD28?), etc. Justify not using an Fc control as FcReceptors transduce signals. If no isotype control was used, say this as a weakness of study.	In line with the Editor's feedback and to streamline the focus of our manuscript, we have opted to omit the proteomics section. We concur with the reviewer that the absence of an Fc control is a limitation of our study. This oversight will be rectified in subsequent experiments, which we plan to present in a different context.	

31. In Supp Fig 1, show data as raw histograms of isotype control and stained cells.	We appreciate the reviewer's suggestion to enhance the clarity of our data presentation. In the revised Supplementary Fig. 13 we have now included raw histograms displaying both the isotype control and stained cells. This representation provides a clearer comparison and underscores the specificity of our staining.	
32. Line 262. Discuss/reference why this is an M1-like set of cytokines.	Macrophages exhibit a spectrum of polarization phenotypes, with M1 and M2 representing the two polar extremes characterized by distinct functions. M1-like macrophages, often termed as 'proinflammatory', generally possess anti-tumor activities. They are typically associated with the secretion of cytokines such as TNF- α , IL-6, IL-12, IL-23, and chemokines like CXCL9, CXCL10, and CXCL11. In contrast, M2-like macrophages, which are considered 'anti-inflammatory' or 'pro-tumoral', are characterized by the production of TGF- β , CCL22, CCL24, and IL-10, among others. Our cytokine profiling indicates an upregulation of many of these signature M1 cytokines and chemokines upon treatment with SNS-101 combined with anti-PD-1. Simultaneously, we observed a decline in several M2-associated immunomodulators. Moreover, the reductions in CCL2, CCL7, and CCL12 levels suggest a potential decrease in factors that drive the differentiation and infiltration of pro-tumor, M2-like tumor-associated macrophages (TAMs) (see Ozga, et al. Chemokines and the immune response to cancer. Immunity 54, 859-874 (2021), now referenced for review).	439-448
33. Line 275. "Among all enriched pathways, 76 (14.4%) correspond to previous findings from functional and transcriptomic studies?" from VISTA studies? Reference	In light of the feedback from the Editor and to sharpen the focus of our manuscript, we have opted to omit the proteomics section. Nonetheless, to address the reviewer's question directly: yes, the mentioned 14.4% overlap pertains to findings from previous VISTA studies. Our original Supplementary Table 4, tab 4, underscored the overlap of proteins we identified in CD4 ⁺ T-cells that were altered upon VISTA-Fc exposure. These proteins corresponded with pathways enriched in a specific VISTA-targeted transcriptomics analysis, as detailed in Yuan et al., Mol Immunol (2023) 157:101-111. [doi: 10.1016/j.molimm.2023.03.021]. Additionally, the overlap was consistent with findings from several other studies, which are referenced below:  • PMID: 37004501 • PMID: 31340983 • PMID: 29375120 • PMID: 24691993 • PMID: 33495077 	
34. The emphasis on IL-7 and TSLP pathways is poorly justified. They are presented as if they	In line with the Editor's feedback and to streamline the narrative of our manuscript, we have opted to omit the proteomics section. We are	

are for sure the important pathways. Other pathways, higher on the list, such as downstream TCR signaling, would seem more likely to be important. Soften the emphasis on IL-7 and TSLP pathways unless a stronger justification can be provided, note that they need experimental validation.	grateful to the reviewer for their astute observations. We acknowledge the critical importance of TCR signaling in response to VISTA binding, which is underscored by its significant enrichment ($p < 0.05$) across multiple signaling databases, namely KEGG, PID BIOCARTA, NETPATH, REACTOME, and PID NCI (as detailed in the Supplementary Table-Statistics and Pathway Analysis). Our pathway analysis, specifically within the NETPATH pathway database, highlighted TSLP and IL-7 as VISTA-altered pathways. As delineated in the main text, TSLP has been documented to synergistically operate with the IL-7 pathway and interfaces with other enriched VISTA-altered pathways, acting as a central hub. Furthermore, TSLP has been shown to bolster CD4+ T-cell homeostasis. Collectively, these findings hint at a potential interplay between VISTA binding and these two pathways, underscoring the need for subsequent experiments to validate their functional significance.	
35. Does the proteomics analysis include phospho-peptides?	In response to the feedback from the Editor and to refine the manuscript's focus, we have removed the proteomics section. To address the reviewer's specific question: our proteomics analysis did not encompass phospho-peptides.	

REVIEWERS' COMMENTS

Reviewer #1 (Remarks to the Author):

Excellent job! Authors addressed all my concerns.

Reviewer #2 (Remarks to the Author):

The authors have satisfactorily addressed my concerns and questions, and the manuscript can be accepted for publication.

Reviewer #3 (Remarks to the Author):

The authors have addressed my concerns.

Thwere were two duplicated sentences.

162. Targeting VISTA with a non-pH-selective antibody led to dose-limiting CRS in patients,

163. 176 ending the clinical trial (NCT02671955)12.

306. The therapeutic

307. 311 potential of VISTA as an immune checkpoint has been challenging to explore due to the

308. 312 previously observed clinical manifestation of CRS12. To circumvent this, we developed

309. 313 SNS-101, a fully human antagonistic VISTA antibody that selectively binds to protonated

310. 314 VISTA at low pH, a critical feature to mitigate CRS.